# Isonicotinylation is a histone mark induced by the anti-tuberculosis first-line drug isoniazid

Yuhan Jiang[1], Yixiao Li[1], Cheng Liu[1], Lei Zhang[1], Danyu Lv[1], Yejing Weng[2], Zhongyi Cheng[2], Xiangmei Chen[3], Jun Zhan[1] & Hongquan Zhang ⬤ [1✉]

Isoniazid (INH) is a first-line anti-tuberculosis drug used for nearly 70 years. However, the mechanism underlying the side effects of INH has remained elusive. Here, we report that INH and its metabolites induce a post-translational modification (PTM) of histones, lysine isonicotinylation ($K_{inic}$), also called 4-picolinylation, in cells and mice. INH promotes the biosynthesis of isonicotinyl-CoA (Inic-CoA), a co-factor of intracellular isonicotinylation. Mass spectrometry reveals 26 $K_{inic}$ sites in histones in HepG2 cells. Acetyltransferases CREB-binding protein (CBP) and P300 catalyse histone $K_{inic}$, while histone deacetylase HDAC3 functions as a deisonicotinylase. Notably, MNase sensitivity assay and RNA-seq analysis show that histone $K_{inic}$ relaxes chromatin structure and promotes gene transcription. INH-mediated histone $K_{inic}$ upregulates *PIK3R1* gene expression and activates the PI3K/Akt/mTOR signalling pathway in liver cancer cells, linking INH to tumourigenicity in the liver. We demonstrate that $K_{inic}$ is a histone acylation mark with a pyridine ring, which may have broad biological effects. Therefore, INH-induced isonicotinylation potentially accounts for the side effects in patients taking INH long-term for anti-tuberculosis therapy, and this modification may increase the risk of cancer in humans.

[1] Program for Cancer and Cell Biology, Department of Human Anatomy, Histology and Embryology, PKU International Cancer Institute, MOE Key Laboratory of Carcinogenesis and Translational Research and State Key Laboratory of Natural and Biomimetic Drugs, Peking University Health Science Center, Beijing, PR China. [2] Jingjie PTM BioLab Co. Ltd., Hangzhou Economic and Technological Development Area, Hangzhou, PR China. [3] Department of Microbiology & Infectious Disease Center, Peking University Health Science Center, Beijing, PR China. ✉email: Hongquan.Zhang@bjmu.edu.cn

Epigenetic mechanisms, including DNA methylation and histone PTMs, possess biomarker potential for tumour classification and prognosis of targeted therapy. Histone PTMs in eukaryotic cells play a crucial role in malignant transformation, tumour development, and progression by regulating chromatin structure and transcriptional activity of genes. Short-chain fatty acids are the primary metabolites of bacteria and cell metabolism. In recent years, a variety of histone acylations have been identified, and these acylations appear to use their corresponding short-chain CoAs also short-chain fatty acids as cofactors and donors, respectively. Cell concentrations of short-chain CoAs and histone acylation levels are affected by short-chain fatty acids[1–5].

The influence of food and drugs on histone PTMs has attracted attention in recent years. For example, aspirin acetylates lysine residues in vivo and in vitro and preservative sodium benzoate benzoylated histones[6–8]. The above evidence suggests that compounds that we consume orally may also have important effects on epigenetic modifications of chromatin.

Tuberculosis (TB) is a global issue, and INH was first reported to be effective for the treatment of TB in 1952[9]. INH remains the first-line drug approved for TB therapy due to its high efficacy. The standard therapies for TB include a combination treatment of INH, rifampicin, pyrazinamide, and ethambutol[10], INH can also be used alone[11]. However, INH is also widely known for its hepatotoxicity and even liver failure risk[12,13]. Several enzymes are involved in INH metabolism and INH metabolites, primarily acetyl-hydrazine (AcHz), hydrazine (Hz), and acetyl-isoniazid (AcINH), are thought to be responsible for the INH-induced liver injury[14–25]. However, the molecular mechanisms underlying the side effects of INH remain unclear.

In this report, we describe the identification and certification of $K_{inic}$ as an INH-induced, evolutionarily conserved PTM in cells. We observed that mice and cellular histone $K_{inic}$ levels were significantly increased after treatment with INH, which promoted the generation of Inic-CoA. Furthermore, this INH-induced modification is dynamically regulated by CBP and P300, two widely used acetyltransferases, and HDAC3, a $Zn^{2+}$-dependent deacetylase. Notably, our study shows that histone $K_{inic}$ relaxes chromatin structures and affects expression of hundreds of genes. Thus, our study identifies a type of histone acylation, which may lead to other investigations of pyridine-containing histones in the epigenetic regulation of gene expression.

## Results

### Identification and characterisation of histone isonicotinylation.

Recently, lysine benzoylation ($K_{bz}$) has been identified as a new type of PTM. $K_{bz}$ is an abundant, evolutionarily conserved, and dynamic histone mark[8]. To identify $K_{bz}$ substrates, we performed a co-immunoprecipitation assay to enrich proteins from HepG2 cells using a pan anti-$K_{bz}$ antibody. Enriched proteins were treated with trypsin and peptides were subjected to HPLC-MS/MS analysis, followed by further analysis using PTMap, a software used for analysing the mass shift caused by PTMs[26]. Surprisingly, we observed a mass shift of +104.0268 Da caused by $K_{bz}$ at low abundance, but we also identified several modified histone peptides, such as KQLATK$_{+105.0215}$AAR, which contained a mass shift of +105.0215 Da at the lysine residue position H3K23 (Supplementary Fig. 1a). The same mass shift of +105.0215 Da was also found in other histone lysine residues, including H3K122, H4K77, and H4K91, as well as non-histone lysine residues including HNRNPC K243 and ACTB K61 (Supplementary Fig. 1b–f). These findings suggest that this mass shift of +105.0215 is likely caused by a modification of histones and non-histones different from $K_{bz}$. Furthermore, these results

indicate that the antibody against $K_{bz}$ could also be used to enrich peptides with structurally related acylations, as previously reported in cases where the acetylation antibody recognises propionylation and butylation[2,3,27–29] (Supplementary Fig. 1g). The most probable elemental composition formula for this modification group was $C_6H_3NO$, based on the accurately determined mass shift. According to the chemical formula, there are three possible lysine modifications that produce the isomers matching $C_6H_3NO$: lysine picolinylation ($K_{pic}$), lysine nicotinylation ($K_{nic}$), and lysine isonicotinylation ($K_{inic}$; Fig. 1a). However, which one exists in cells remained unknown.

To determine which isomer causes the +105.0215 Da mass shift, we first determined whether the pan anti- $K_{bz}$ antibody recognises the above three isomers. We designed and synthesised four peptides based on the histone H4 amino acid sequence containing the lysine with a + 105.0215 Da mass shift: DAVTY-TEHAKR, DAVTYTEHAK$_{pic}$R, DAVTYTEHAK$_{nic}$R, and DAV-TYTEHAK$_{inic}$R. Dot-blot analysis was used to examine the reactivity of the anti-$K_{bz}$ antibody with the four peptides. The results showed that the anti-$K_{bz}$ antibody recognised all three modified peptides in an in vitro experimental setup (Supplementary Fig. 2a), confirming that the anti-$K_{bz}$ antibody could also recognise lysine picolinylation, lysine nicotinylation, and lysine isonicotinylation in vitro. Intriguingly, when we analysed the structures of the three isomers, we found that the structure of $K_{inic}$ is similar to INH. Therefore, we wanted to characterise $K_{inic}$ based on histone modification. To identify this PTM on histones, a customer-designed pan anti-$K_{inic}$ antibody was generated, which showed a high specificity for $K_{inic}$ but not for $K_{pic}$ and $K_{nic}$, as shown by the Dot-blot assay (Fig. 1b). This antibody was able to detect endogenous $K_{inic}$ signals of core histones and non-histones using lysates from HepG2 and HeLa cells in a Western blot assay (Fig. 1c, Supplementary Fig. 2b). Additionally, immunofluorescent analysis using this antibody demonstrated that $K_{inic}$ was primarily distributed in the nuclei of living cells (Fig. 1d).

Using this $K_{inic}$ antibody, we also detected histones and non-histones from insect cells Sf-9 (Spodoptera frugiperda 9), human HepG2, HeLa, HEK293T, HCT-116, HT-29, NCI-H157, NCI-H1299, mouse embryonic fibroblasts MEF cells, human HK-2, MCF-7, SUM159, SW1116, HEK293A, and MCF-10A cells. $K_{inic}$ signals were detected at various levels. These results indicate that $K_{inic}$ is a widely existed modification in eukaryotic cells (Fig. 1e, Supplementary Fig. 2c). Furthermore, we performed immuno-histochemical staining in a variety of human normal tissues and found that in normal thyroid, colon, and lung tissues, the $K_{inic}$ levels were low; however, in normal oesophageal epithelium and fundic gland tissues, the $K_{inic}$ levels were high (Fig. 1f). We also noticed an interesting phenomenon in the normal oesophageal epithelium tissue: the isonicotinylation is higher in the well-differentiated outer layer than in the inner layer (Fig. 1f).

### INH stimulates histone $K_{inic}$ in cells and mice.

Acylations are usually modulated by fatty acid metabolites[1,2,4,30]; for example, acetate induces acetylation[31]. Therefore, we determined whether isonicotinic acid (INA) or sodium isonicotinate (SIN) could induce $K_{inic}$. We applied a series of concentrations of INA or SIN to induce $K_{inic}$ in HepG2, HEK293A, and HCT-116 cells for 24 h. Western blot analysis showed that histone $K_{inic}$ levels were induced in a dose-dependent manner in these cells, and the same induction for $K_{inic}$ was also seen in non-histone proteins. In contrast, no significant change was observed in overall histone or non-histone acetylation (Fig. 2a–c, Supplementary Fig. 3a–j). These findings clearly indicate that $K_{inic}$ can be induced by both intracellular INA and SIN in vivo. Given that INA is a

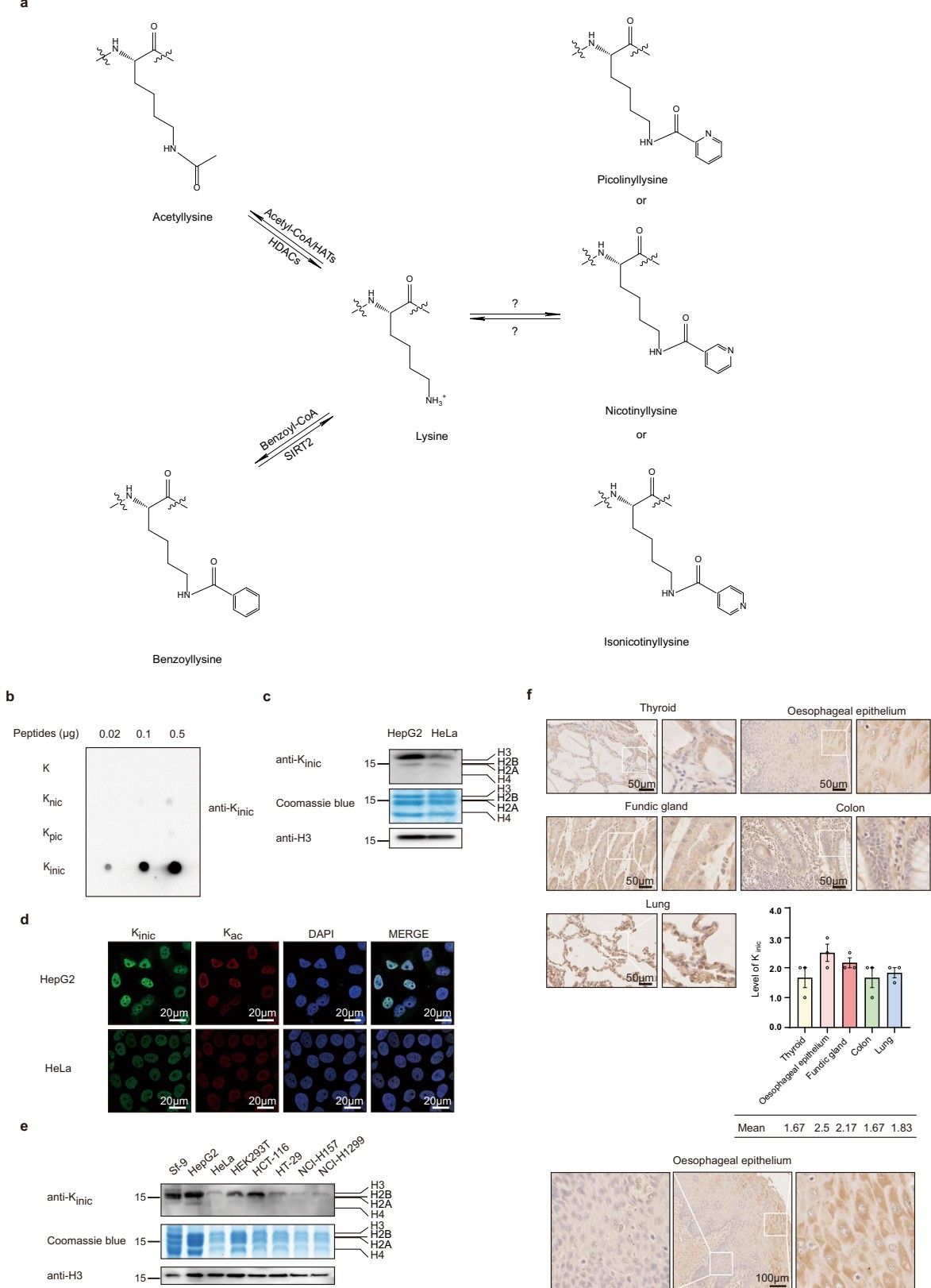

downstream metabolite of INH, we evaluated whether INH can also induce $K_{inic}$ in cells because they share the same structural backbone. To test this hypothesis, we detected histone and non-histone $K_{inic}$ levels in the above three cell lines after increasing the concentration of INH, and the results showed that both histone and non-histone $K_{inic}$ levels were obviously induced by INH

(Fig. 2d, Supplementary Fig. 3k–o). These results were also confirmed by immunofluorescent staining, in which $K_{inic}$ levels in the cell nuclei substantially increased after treatment with INH, INA, and SIN, whereas no significant change was observed in lysine acetylation levels (Fig. 2e, Supplementary Fig. 3p). To examine whether INH could induce $K_{inic}$ in animals, we administered

**Fig. 1 Identification and characterization of histone isonicotinylation. a** Chemical structures of lysine acetylation, lysine benzoylation and three structural isomers modifications (lysine picolinylation, lysine nicotinylation and lysine isonicotinylation) that have the elemental compositions of $C_6H_3NO$ and may induce a mass shift of +105.0215 Da. **b** Dot-blot assay was used to verify the specificity of pan antibody against $K_{inic}$ modification. Nitrocellulose membrane was spotted by different amounts of unmodified peptide (DAVTYTEHAKR), lysine nicotinylated peptide (DAVTYTEHAK$_{nic}$R), lysine picolinylated peptide (DAVTYTEHAK$_{pic}$R), lysine isonicotinylated peptide (DAVTYTEHAK$_{inic}$R) and detected with the pan-$K_{inic}$ antibody. **c** Detection of histone $K_{inic}$ marks in HepG2 and HeLa cells. Core histones acid-extracted from HepG2 and HeLa cells was tested using pan-$K_{inic}$, pan-$K_{ac}$, and H3 antibodies by Western blot. Total histones were visualized with Coomassie blue staining. **d** Subcellular localization of $K_{inic}$ marks is mainly in nucleus. HepG2 and HeLa cells were stained with pan-$K_{inic}$ rabbit (green) and pan-$K_{ac}$ mouse (red) antibodies. Nuclei were stained with DAPI (blue), followed by visualization with confocal microscopy. Scale bar, 20 μm. **e** Detection of histone $K_{inic}$ marks in Sf-9, HepG2, HeLa, HEK293T, HCT-116, HT-29, NCI-H157, NCI-H1299 cells. Core histones acid-extracted from those cells were tested using pan-$K_{inic}$, pan-$K_{ac}$, and H3 antibodies by Western blot. Total histones were visualized with Coomassie blue staining. **f** Detection of $K_{inic}$ marks in normal tissues. Human normal thyroid, oesophageal epithelium, fundic gland, colon, and lung tissues were performed by immunohistochemical staining with pan-$K_{inic}$ antibody respectively, followed by visualization with light microscopy. Scale bar, 50 μm, and quantifications of the scores of immunohistochemical staining (upper, $n = 3$ samples, values were expressed as mean ± SEM). Detection of $K_{inic}$ marks in oesophageal carcinoma tissue. Scale bar, 100 μm (bottom).

gavage feeding mice with 50 mg/kg/day of INH or normal saline as a control for ten days. The regular dose of INH during TB therapy is 5 mg/kg/day in humans and the 50 mg/kg/day INH dose in mice mimics the pharmacological dose in human[13]. Interestingly, we found that both histone and non-histone $K_{inic}$ levels were increased in mouse liver tissues after INH treatment (Fig. 2f). This result indicated that INH could induce $K_{inic}$ in mice and maybe also do so in humans who take INH for TB therapy.

It is known that if two peptides have the same MS/MS fragmentation patterns and approximately identical retention times, they are considered to have an identical structure. Therefore, to scrutinise whether the INH-induced PTM in vivo is the genuine $K_{inic}$, we analysed in vivo peptide and the in vitro synthetic peptide DAVTYTEHAK$_{inic}$R by HPLC-MS/MS. Results showed that they have the same MS/MS fragmentation patterns and almost identical retention times (21.50 min *vs.* 21.49 min; Supplementary Fig. 3q–r). Therefore, we concluded that there are peptides DAVTYTEHAK$_{inic}$R in vivo with $K_{inic}$, indicating that the INH-induced PTM in cells is $K_{inic}$.

Next, we investigated whether INH-induced $K_{inic}$ is caused by INH directly or by its metabolite, INA. It was known that INH can be converted into INA by two enzymes, N-acetyltransferase 2 (NAT2) and isoniazid hydrolase[32]. As shown in the left panel of Fig. 2g, we separately blocked intracellular INH metabolism through NAT2 inhibitors 1-aminobenzotriazole (ABT) and acetaminophen (APAP), two newly identified NAT2 inhibitors[33,34], and isoniazid hydrolase inhibitor bis-p-nitrophenyl phosphate (BNPP)[35]. Results showed that when INH metabolism was separately blocked, the intracellular levels of histone $K_{inic}$ were not affected (Fig. 2g, right panel), indicating that INH likely induces histone $K_{inic}$ directly, without the requirement of INH turnover into INA. Furthermore, we found that histone $K_{inic}$ levels reached a steady state after INH treatment for 12 h, while there was no change in histone acetylation levels (Supplementary Fig. 3s).

**INH is a donor for generating intracellular isonicotinyl-CoA.** Accumulating evidence has shown that acylation donors stimulate the production of the corresponding cellular acyl-CoAs, resulting in elevated levels of histone acylations[3,5,8,36,37]. Although there is no data on Inic-CoA, it is the most likely co-factor for the $K_{inic}$ reaction in cells, similar to acetyl-CoA for the lysine acetylation reaction. To determine whether the Inic-CoA exists intracellularly, we chemically synthesised Inic-CoA as a control, based on the chemical formula shown in Supplementary Fig. 3t. We metabolically labelled HepG2 cells with 5 mM isotopic D4-INH for 24 h, followed by mass spectrometry-based quantification. The results showed that there was an obviously increase in D4-Inic-CoA upon D4-INH treatment (Fig. 2h, right panel),

while no change was observed for Ac-CoA as a control (Fig. 2h, left panel). Moreover, after treatment with increasing concentrations of INH, the HPLC-MS/MS measurements showed an elevated cellular concentration of Inic-CoA, which was dose-dependent (Fig. 2i). Overall, we showed the existence of Inic-CoA in living cells and demonstrated the histone mark, $K_{inic}$, with the pyridine ring, which is initiated by INH-stimulated production of Inic-CoA.

**Identification of $K_{inic}$ sites in histones of HepG2 cells.** It is important to identify all $K_{inic}$ sites in histones and compare them with the other histone marks. To this end, HepG2 cells were treated with 10 mM INH for 24 h, and the histones were purified and subjected to trypsin digestion. Peptides were enriched by the pan anti-$K_{inic}$ antibody and analysed by HPLC-MS/MS analysis, followed by a protein sequence database search. Results showed that a total of 26 unique histone $K_{inic}$ sites were identified (Fig. 3). Notably, the histone $K_{inic}$ sites in HepG2 cells are widely distributed, resembling the distribution of histone $K_{ac}$. In contrast, histone $K_{bz}$ sites are primarily located at the N-terminal tails of histones, according to a previous report[8]. These findings indicate that $K_{inic}$, as a histone mark, may play a significant role in the regulation of gene expression alone or in cooperation with other histone modifications.

**CBP/P300 function as isonicotinyltransferases and HDAC3 as deisonicotinylase for histone $K_{inic}$.** Given that we identified the donor and co-factor for the generation of histone isonicotinylation, we attempted to find the transferases that transfer the iso-nicotinyl group to lysine on histones. Based on previous studies on histone acylation modifications, the histone acylations and deacylations are regulated commonly by histone acetyl-transferases (HATs) and deacetylases (HDACs) in vivo. We transfected HepG2 cells with four HAT plasmids, including CBP, P300, PCAF, and hMOF, and found that the histones were obviously isonicotinylated by CBP and P300 but weakly iso-nicotinylated by PCAF and hMOF (Fig. 4a). When HepG2 and HeLa cells were transfected with CBP and P300, histone iso-nicotinylation and acetylation levels increased (Fig. 4b, Supplementary Fig. 4a). Moreover, when endogenous CBP and P300 were knocked down by siRNA or inhibited by A-485, a newly reported inhibitor of CBP/P300[38], and SGC-CBP30, a widely used inhibitor of CBP/P300[39], the levels of histone isonicotinylation and acetylation were decreased, indicating that CBP and P300 are responsible for the "writing" during histone isonicotinylation (Supplementary Fig. 4b–c). In addition, an in vitro iso-nicotinylation reaction experiment was performed to confirm the direct role of CBP/P300 in isonicotinylating histones in the

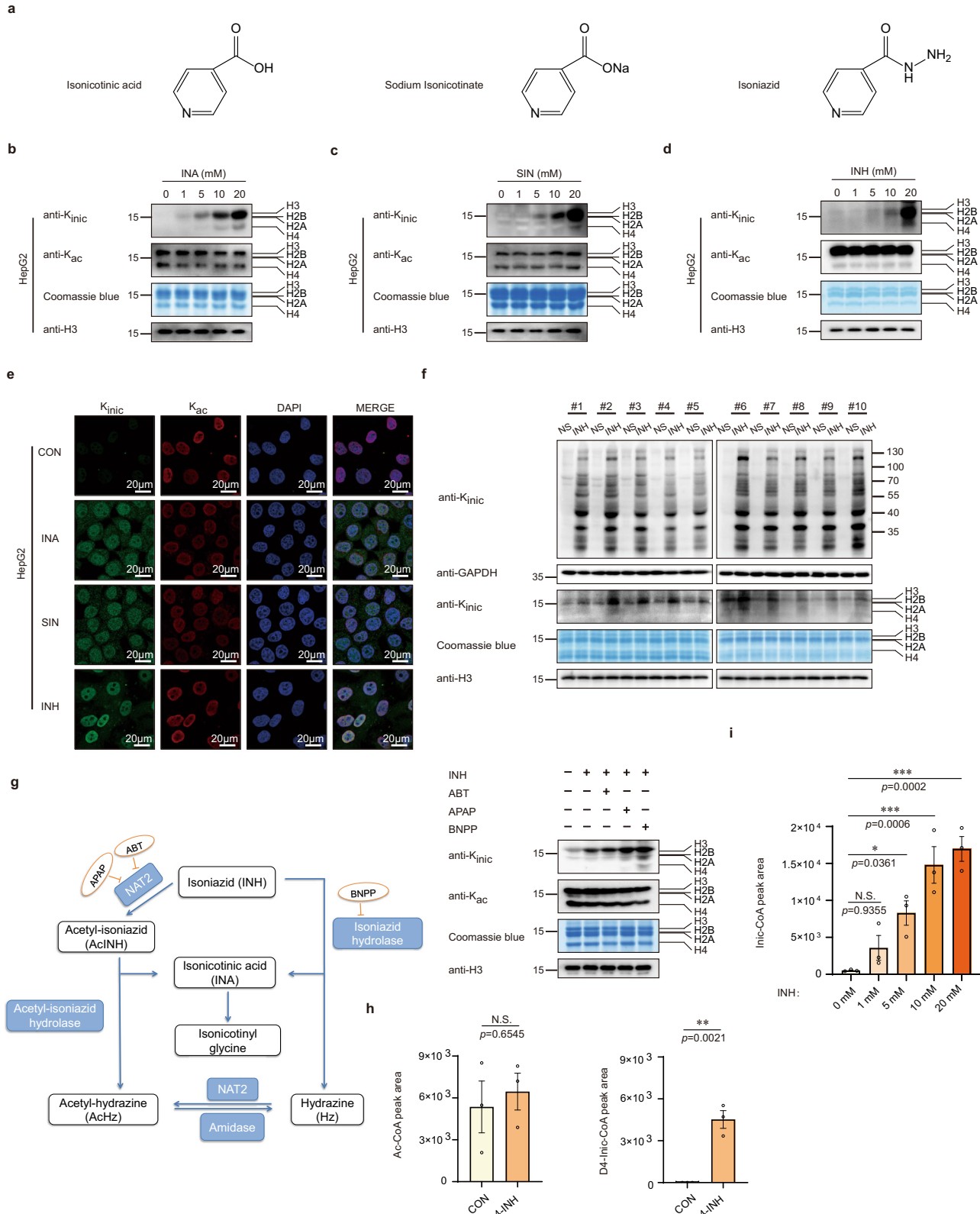

presence of synthetic Inic-CoA and purified CBP/P300. The results showed that CBP/P300 potently isonicotinylated histone $K_{inic}$ in vitro (Fig. 4c).

Next, HepG2 cells were treated with trichostatin A (TSA), a broad inhibitor of HDAC family deacetylases, or nicotinamide (NAM), an inhibitor of SIRT family deacetylases. We found that histone isonicotinylation levels increased after TSA treatment,

while no change was observed after NAM treatment, indicating that HDAC family members influence histone isonicotinylation (Fig. 4d). HepG2 cells were transfected with a variety of expression vectors encoding HDAC family members, including HDAC1-9, and when HepG2 cells were transfected with only HDAC3, the isonicotinylation level obviously decreased, as measured by immunofluorescent staining, but not with other

**Fig. 2 INH stimulates histone K$_{inic}$ in cells and mice. a–e** INA, SIN, and INH stimulate HepG2 histone K$_{inic}$. **a** Chemical structures of isonicotinic acid (INA), sodium isonicotinate (SIN), and isoniazid (INH). **b** Core histones acid-extracted from HepG2 cells that treated with increasing concentration of INA for 24 h and tested using pan-K$_{inic}$, pan-K$_{ac}$, and H3 antibodies by Western blot analysis. Total histones were visualized with Coomassie blue staining. "INA" indicates isonicotinic acid. **c** Core histones acid-extracted from HepG2 cells that treated with increasing concentration of SIN for 24 h and tested using pan-K$_{inic}$, pan-K$_{ac}$, and H3 antibodies by Western blot analysis. Total histones were visualized with Coomassie blue staining, "SIN" indicates sodium isonicotinate. **d** Core histones acid-extracted from HepG2 cells that treated with increasing concentration of INH for 24 h and tested using pan-K$_{inic}$, pan-K$_{ac}$, and H3 antibodies by Western blot analysis. Total histones were visualized with Coomassie blue staining, "INH" indicates Isoniazid. **e** HepG2 cells were treated with 10 mM INA, SIN, and INH respectively for 24 h and stained with pan-K$_{inic}$ rabbit (green) and pan-K$_{ac}$ mouse (red) antibodies. Nuclei were stained with DAPI (blue), followed by visualization with confocal microscopy. Scale bar, 20 μm. **f** INH stimulates mice liver K$_{inic}$ levels. Liver samples from mice treated with normal saline (NS) or 50 mg/kg/day of INH for ten days were collected (*n* = 10 mice each group). And core histones and non-histones were extracted and tested using pan-K$_{inic}$, GAPDH and H3 antibodies by Western blot. Total histones were visualized with Coomassie blue staining. "NS" indicates normal saline treated mice, "INH" indicates isoniazid treated mice. **g** INH is like to stimulate histone K$_{inic}$ directly. Schematic diagram of INH metabolism in vivo (left), core histones acid-extracted from normal HepG2 cells, 1 mM ABT (1-Aminobenzotriazole), 5 mM APAP (Acetaminophen) or 15 mM BNPP (Bis-p-nitrophenyl phosphate) pretreated HepG2 cells respectively, and also treated with 10 mM INH for 24 h, then tested using pan-K$_{inic}$, pan-K$_{ac}$ and H3 antibodies by Western blot analysis. Total histones were visualized with Coomassie blue staining (right). **h–i** INH stimulates the production of Inic-CoA. **h** CoA was extracted from HepG2 cells treated with or without D4-INH (PH = 7.4) for 24 h, then tested the Ac-CoA (left) and D4-Inic-CoA (right) levels using HPLC-MS/MS assay, "CON" indicates untreated HepG2 cells; "D4-INH" indicates D4-isoniazid treated HepG2 cells, quantifications of the chromatographic peak area, data presented are the mean ± SEM from three biological replicates (*n* = 3), as determined by unpaired two-tailed Student's *t*-test. **i** CoA was extracted from HepG2 cells treated with increasing concentration of INH (PH = 7.4) for 24 h, and tested the Inic-CoA levels using HPLC-MS/MS assay, quantifications of the chromatographic peak area, data presented are the mean ± SEM from three biological replicates (*n* = 3), as determined by one-way ANOVA followed by Bonferrroni's multiple comparisons test.

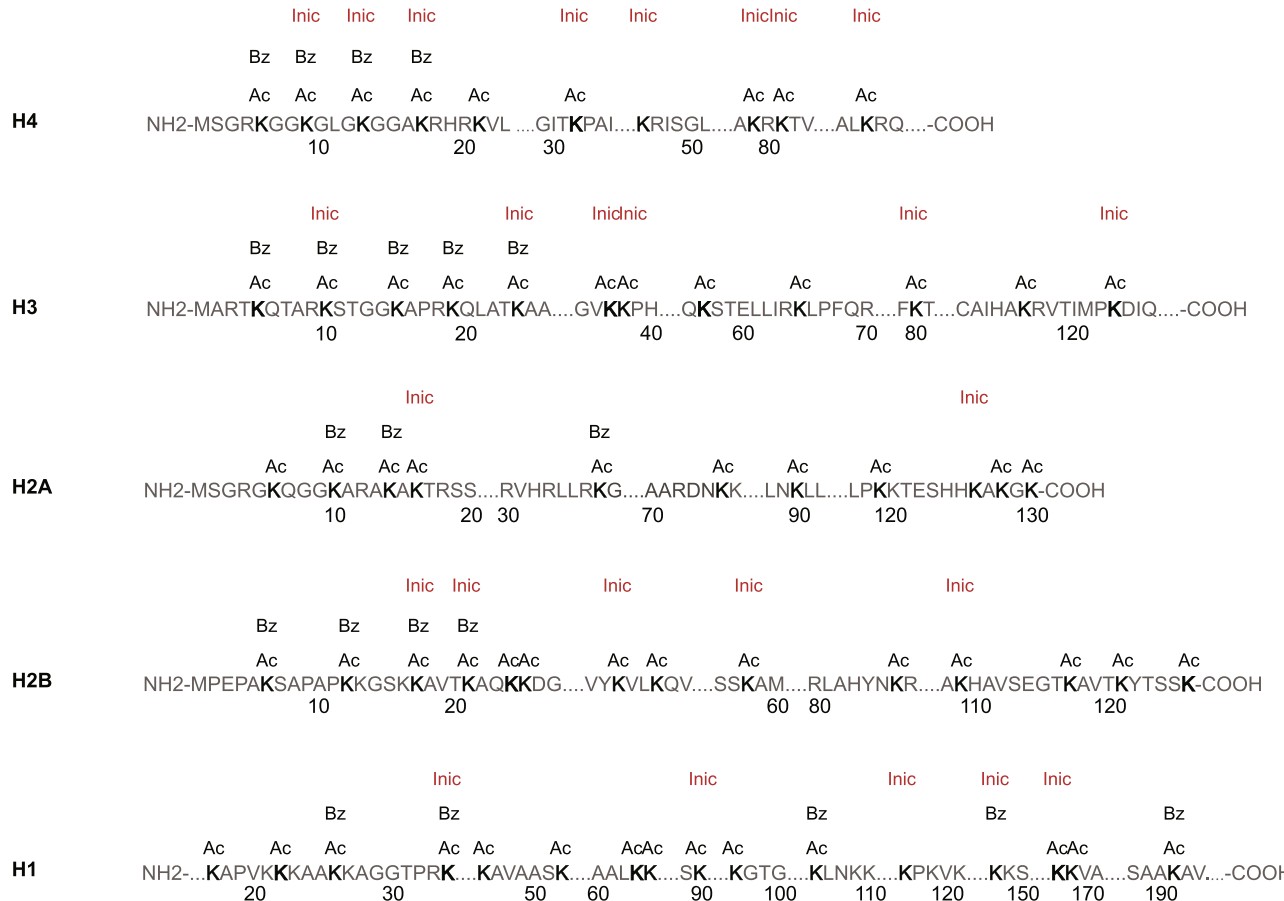

**Fig. 3 Identification of K$_{inic}$ sites in histones of HepG2 cells.** The detected K$_{inic}$ sites in HepG2 cells treated with 10 mM INH for 24 h using HPLC-MS/MS assay are shown in red. And histone K$_{ac}$ and K$_{bz}$ sites that were reported previously are shown for comparison. "K$_{ac}$" indicates lysine acetylation, "K$_{bz}$" indicates lysine benzoylation, "K$_{inic}$" indicates lysine isonicotinylation.

HDAC family members (Fig. 4e, Supplementary Fig. 4d). Furthermore, the Western blot results showed that HDAC3 markedly deisonicotinylates histones among the HDAC family members (Fig. 4f, Supplementary Fig. 4e). When endogenous HDAC3 was knocked down by siRNA, the levels of histone isonicotinylation and acetylation were increased (Supplementary Fig. 4f). Based on the above results, we believe that HDAC3 is a histone deisonicotinylase. Collectively, these results indicate that CBP and P300 act as the histone isonicotinyltransferases, and deacetylase HDAC3 acts as histone deisonicotinylase.

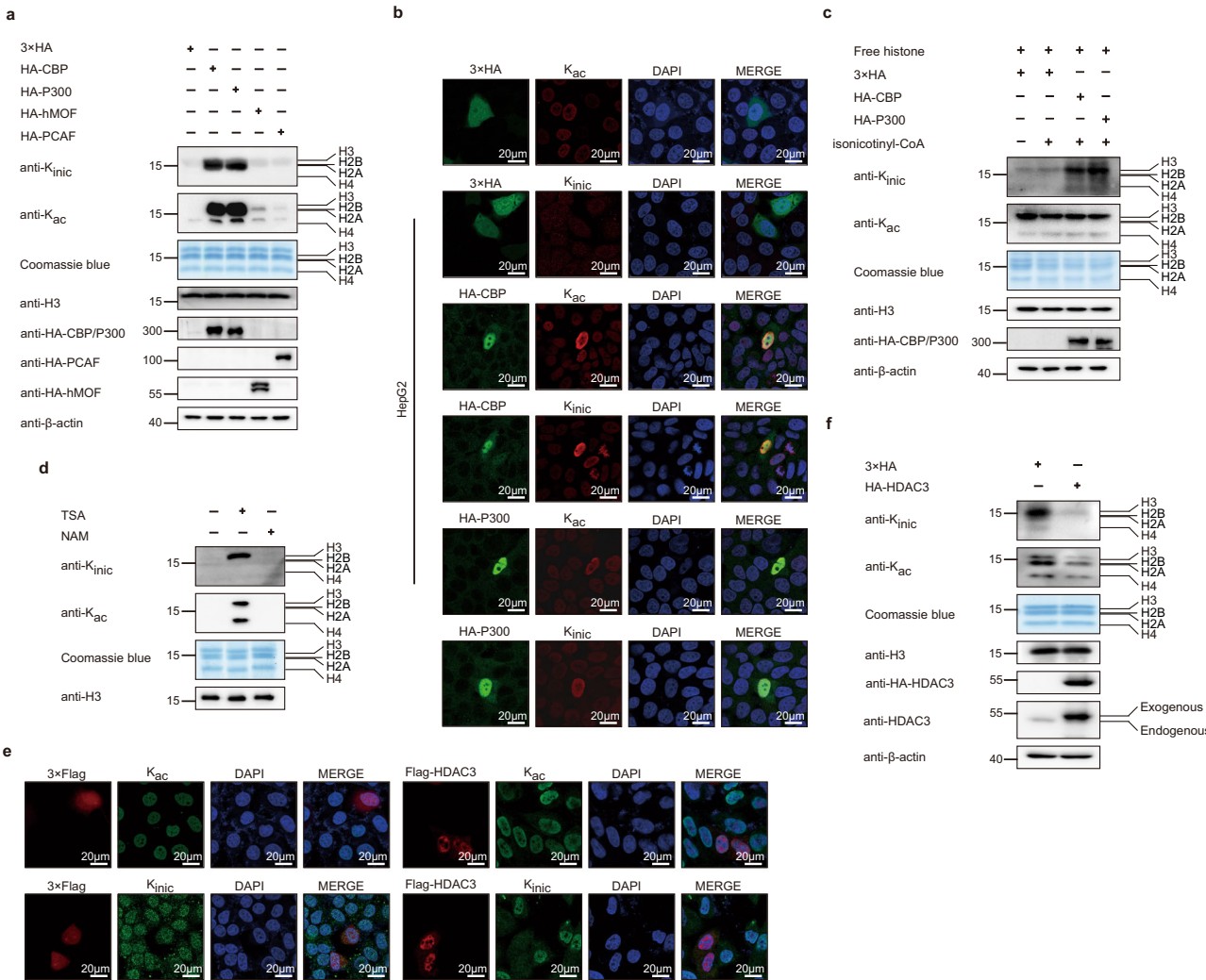

**Fig. 4 CBP/P300 function as isonicotinyltransferases and HADC3 as deisonicotinylase for histone K$_{inic}$.** **a–c** CBP/P300 catalyze histone K$_{inic}$ in vivo and in vitro. **a** HepG2 cells were transfected with 3×HA, HA-CBP, HA-P300, HA-hMOF and HA-PCAF plasmids respectively, core histones acid-extracted and tested using pan-K$_{inic}$, pan-K$_{ac}$ and H3 antibodies. Total histones were visualized with Coomassie blue staining. And total proteins were extracted and tested using HA-tag and β-actin antibodies by Western blot analysis. **b** HepG2 cells were transfected with 3×HA, HA-CBP, or HA-P300 and stained with pan-K$_{ac}$ or pan- K$_{inic}$ rabbit (red) and HA-tag mouse (green) antibodies. Nuclei were stained with DAPI (blue), followed by visualization with confocal microscopy. Scale bar, 20 μm. **c** HA-CBP and HA-P300 plasmids were transfected into HEK293T cells, then the protein was purified with HA beads and subjected to in vitro isoniconitylation assay with or without Inic-CoA using free core histones from HepG2 cells as substrates. Tested using pan-K$_{inic}$, pan-K$_{ac}$, and H3 antibodies. Total histones were visualized with Coomassie blue staining. And total proteins were extracted and tested using HA-tag and β-actin antibodies by Western blot analysis. **d–f** HDAC3 removes histone K$_{inic}$. **d** Core histones acid-extracted from HepG2 cells that treated with 3 μM TSA (Trichostatin A) or 5 mM NAM (Nicotinamide) for 12 h and tested using pan-K$_{inic}$, pan-K$_{ac}$, and H3 antibodies by Western blot. Total histones were visualized with Coomassie blue staining. **e** HepG2 cells were transfected with 3×Flag or Flag-HDAC3 plasmid and stained with pan-K$_{ac}$ or pan-K$_{inic}$ rabbit (green) and Flag-tag mouse (red) antibodies. Nuclei were stained with DAPI (blue), followed by visualization with confocal microscopy. Scale bar, 20 μm. **f** HepG2 cells were transfected with 3×HA, HA-HDAC3 plasmids, core histones acid-extracted and tested using pan-K$_{inic}$, pan-K$_{ac}$, and H3 antibodies. Total histones were visualized with Coomassie blue staining. Total proteins were extracted and tested using HA-tag, HDAC3, and β-actin antibodies by Western blot analysis.

**Histone K$_{inic}$ leads to higher chromatin accessibility and promotes gene transcription.** It is known that histone acylations can influence nucleosome configuration and chromatin accessibility; therefore, we investigated the role of isonicotinylation in the regulation of chromatin accessibility using an MNase sensitivity assay. The results showed that INH-induced histone isonicotinylation leads to higher chromatin accessibility by displaying nucleosomal DNA sensitivity to MNase treatment in HepG2 cells (Fig. 5a). In addition, Sat-2 is a marker for alteration of the heterochromatin structure, and its transcription is generally repressed. However, when the centromeric region is relaxed, Sat-2 gene transcription is increased[40,41]. Sat-2 gene expression during

INH induction was determined. The results showed that Sat-2 gene expression was upregulated as determined by real-time RT-PCR assay in HepG2 cells, indicating that INH-induced histone isonicotinylation also leads to heterochromatin relaxation in HepG2 cells (Fig. 5b). Furthermore, to determine which genes are potentially regulated by histone K$_{inic}$ and which physiological functions may be linked to histone K$_{inic}$, we performed RNA-seq for total RNAs from INH-treated HepG2 cells. In this analysis, 313 genes were found to be upregulated, whereas 121 genes were downregulated (Fig. 5c). The subsequent KEGG pathway enrichment analysis showed that the genes upregulated by histone isonicotinylation were primarily correlated with the TNF

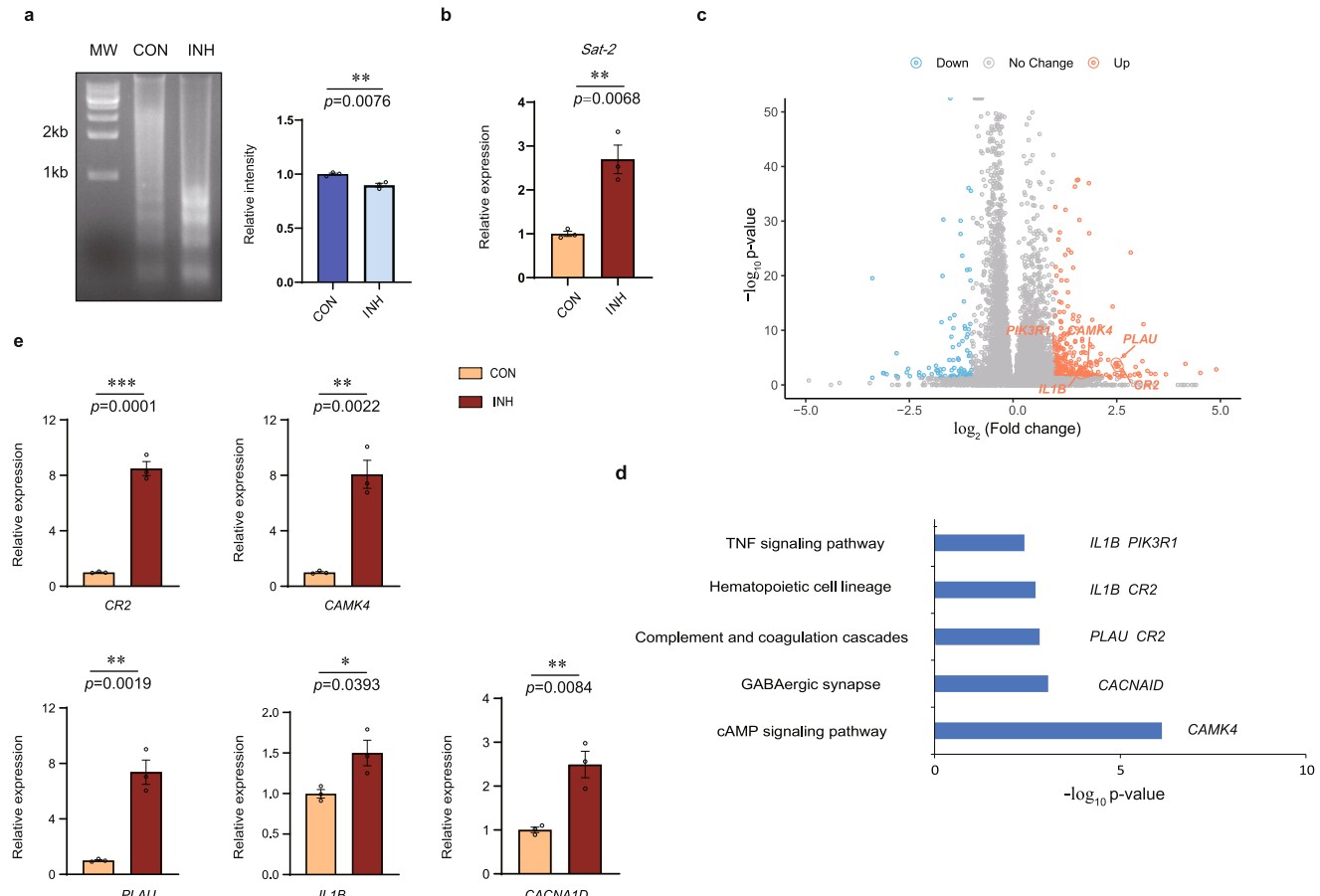

**Fig. 5 Histone K$_{\text{inic}}$ leads to higher chromatin accessibility and promotes gene transcription. a, b** Histone K$_{\text{inic}}$ induces chromatin accessibility. **a** HepG2 cells were treated with or without 10 mM INH for 6 h, nucleosomal DNA was extracted and analyzed by MNase sensitivity assay (left). "CON" indicates untreated HepG2 cells; "INH" indicates isoniazid treated HepG2 cells, MW: Molecular Weight. Quantifications of lane signal intensity (right, upper bands), data presented are the mean ± SEM from three biological replicates ($n = 3$), as determined by unpaired two-tailed Student's $t$-test **b** Relative mRNA expression of *Sat-2* in HepG2 cells treated with or without 10 mM INH for 12 h and determined by real-time RT-PCR assay, "CON" indicates untreated HepG2 cells; "INH" indicates isoniazid treated HepG2 cells, data presented are the mean ± SEM from three biological replicates ($n = 3$), as determined by unpaired two-tailed Student's $t$-test. **c** Histone K$_{\text{inic}}$ upregulates 313 genes and downregulates 121 genes. Volcano plot analysis of pairwise comparison of RNA-seq results from HepG2 cells with or without 10 mM INH for 12 h ($n = 3$ samples, values were expressed as mean ± SEM). **d** KEGG pathway analysis of mRNA levels elevated genes. Some genes involved in each pathway are labelled. **e** Real-time RT-PCR analysis of the genes labelled in KEGG pathway analysis. Relative expression is normalized to GAPDH, "CON" indicates untreated HepG2 cells; "INH" indicates isoniazid treated HepG2 cells, data presented are the mean ± SEM from three biological replicates ($n = 3$), as determined by unpaired two-tailed Student's $t$-test.

signalling pathway, hematopoietic cell lineage, complement and coagulation cascades, GABAergic synapse, and cAMP signalling pathway (Fig. 5d). Among these pathways, the TNF signalling pathway is closely related to inflammation and cell death. Next, we chose five genes to validate the correlation between histone isonicotinylation and gene expression using real-time RT-PCR assays: *CR2, CAMK, PLAU, IL1B,* and *CACNA1D*. Real-time RT-PCR results showed that the mRNA expression of these selected genes was upregulated by 1.50- to 8.49-fold (*CR2*: 8.49-fold, *CAMK*: 8.08-fold, *PLAU*: 7.37-fold, *IL1B*: 1.50-fold, *CACNA1D*: 2.50-fold; Fig. 5e). These findings indicate that histone isonicotinylation, like other acylations, promotes gene transcription by loosening the chromatin structure in the genome.

**INH-induced histone isonicotinylation varies in tumours and upregulated cancer-related signalling pathway.** INH is a very effective first-line anti-TB drug. However, it is known that patients who take INH to cure TB suffered side effects. Thus, many investigations were carried out in the 1970s with a focus on the toxicity of INH[42]. INH is known to induce liver cancer in

rats[43], but it is unknown whether INH can cause cancer in humans. To examine whether INH-induced isonicotinylation is correlated with cancer in humans, we collected ten different kinds of tumour tissues and normal tissues adjacent to tumours and used immunohistochemical staining to determine the levels of isonicotinylation in these tissues. Results showed that the total K$_{\text{inic}}$ level was lower in gastric carcinoma tissues than in normal tissues, while it was higher in lung adenocarcinoma and thyroid carcinoma tissues than in normal tissues. However, the total K$_{\text{inic}}$ level in renal carcinoma, rectal carcinoma, colonic adenocarcinoma, mammary adenocarcinoma, pancreatic carcinoma, oesophageal carcinoma, and hepatocellular carcinoma tissues were not significantly different from those in normal tissues. (Fig. 6a, b, Supplementary Fig. 5a). These findings suggest that isonicotinylation levels may be linked to certain types of cancer. Therefore, isonicotinylation of a particular protein, either histone or non-histone proteins, might be correlated with cancer development.

Intriguingly, in HepG2 cells treated with INH, RNA-seq analysis revealed that the *PIK3R1* gene was upregulated. This gene encodes the protein PIK3R1 (also called P85α), which is the

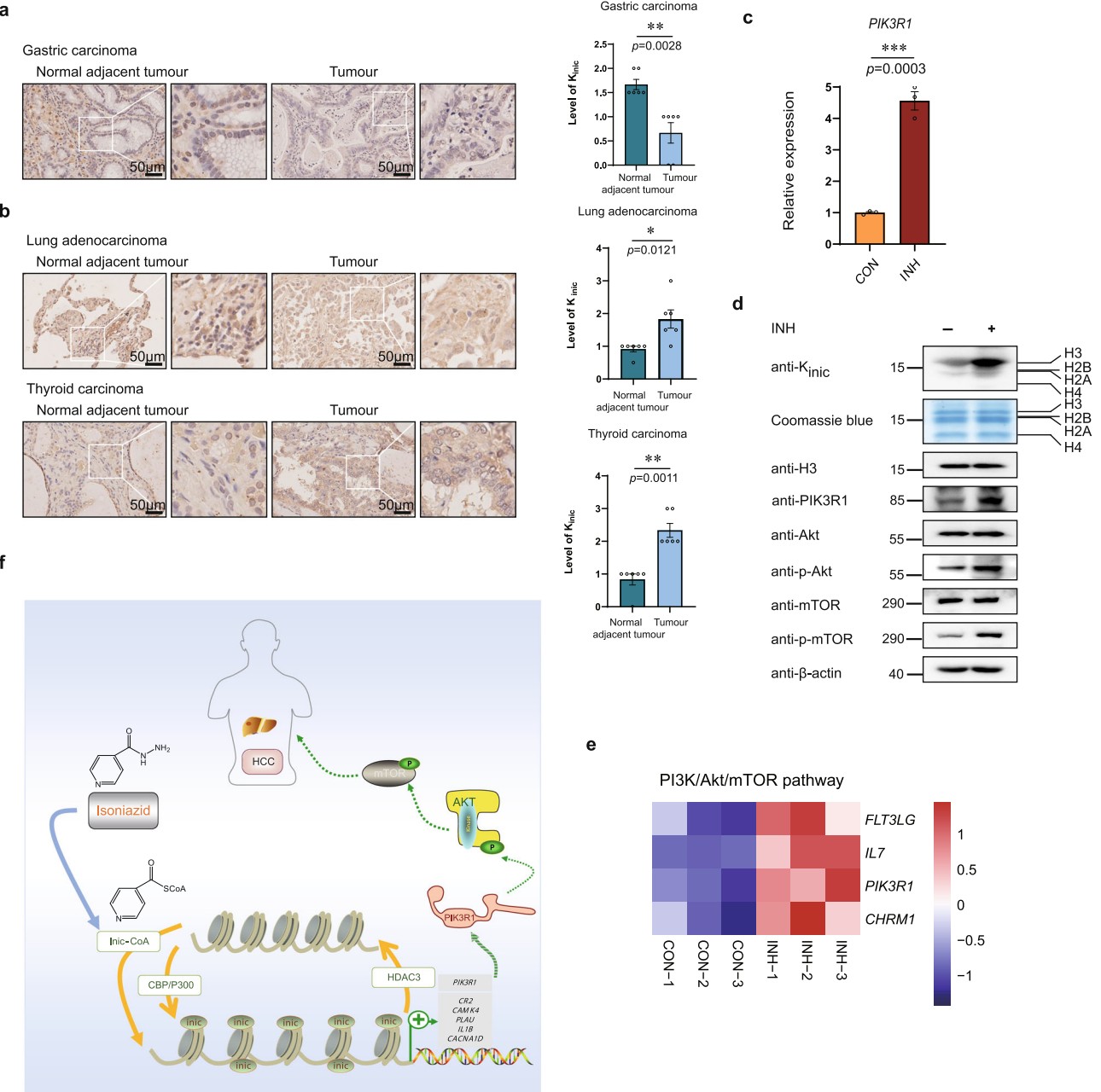

**Fig. 6 INH-induced histone isonicotinylation varies in tumours and upregulated cancer-related signalling pathway. a, b** Detection of K$_{inic}$ mark in normal adjacent tumour tissues and tumour tissues. **a** Normal adjacent tumour tissue and tumour tissue of gastric carcinoma were performed by immunohistochemical staining with pan-K$_{inic}$ antibody, followed by visualization with light microscopy. Scale bar, 50 μm (left). Quantifications of the scores of immunohistochemical staining (right) ($n = 6$ samples, values were expressed as mean ± SEM, as determined by paired two-tailed Student's $t$-test). **b** Normal adjacent tumour tissues and tumour tissues of lung adenocarcinoma and thyroid carcinoma were performed by immunohistochemical staining with pan-K$_{inic}$ antibody respectively, followed by visualization with light microscopy. Scale bar, 50 μm. Quantifications of the scores of immunohistochemical staining (right) ($n = 6$ samples, values were expressed as mean ± SEM, as determined by paired two-tailed Student's $t$-test). **c** Relative mRNA expression of *PIK3R1* in HepG2 cells treated with or without 10 mM INH for 12 h and determined by real-time RT-PCR assay. Relative expression is normalized to GAPDH, "CON" indicates untreated HepG2 cells; "INH" indicates isoniazid treated HepG2 cells, data presented are the mean ± SEM from three biological replicates ($n = 3$), as determined by unpaired two-tailed Student's $t$-test. **d** HepG2 cells were treated with or without 10 mM INH for 18 h, core histones acid-extracted and tested using pan-K$_{inic}$ and H3 antibodies. Total histones were visualized with Coomassie blue staining. And total proteins were extracted and tested using PIK3R1, Akt, p-Akt, mTOR, p-mTOR, and β-actin antibodies by Western blot analysis. **e** Heatmap displaying differentially expressed genes involved in PI3K/Akt/mTOR pathway in response to INH in RNA-seq data. "CON" indicates untreated HepG2 cells; "INH" indicates isoniazid treated HepG2 cells; $p < 0.05$, log FC > 1. **f** A graphical model of histone isonicotinylation.

regulatory subunit of PI3K and was previously reported to promote HCC progression via the PI3K/Akt/mTOR pathway[44–46]. We found that the mRNA and protein levels of PIK3R1 were greatly increased when histones were isonicotinylated by INH stimulation. Notably, the levels of p-Akt and p-mTOR also increased, indicating that the mTOR signalling pathway was activated by histone $K_{inic}$ (Fig. 6c, d). In addition, we enriched PI3K/Akt/mTOR signalling pathway enrichment based on the results of RNA-seq analysis and found that histone isonicotinylation promotes the expression of multiple genes of the PI3K/Akt/mTOR pathway, including *FLT3LG*, *IL7*, *PIK3R*, and *CHRM1* (Fig. 6e, Supplementary Fig. 5b).

This finding demonstrated that INH-induced histone isonicotinylation activates the PI3K/Akt/mTOR signalling pathway, which maybe the evidence of histone isonicotinylation linked to cancer development. Subsequently, we randomly selected ten paired HCC patient samples and detected the profile of total isonicotinylation in tumours and adjacent normal tissues. The results showed that two of the ten patients (patients #3 and #6) displayed higher levels of isonicotinylation in tumour tissues than in the adjacent normal tissues, and the levels of PIK3R1 were increased concomitantly in tumour tissues of the same patients (patients #3 and #6; Supplementary Fig. 5c).

## Discussion

In the present study, we evaluate the histone mark $K_{inic}$, a type of acylation. This histone modification is stimulated by INH, the first-line anti-tuberculosis drug discovered 70 years ago that is still effective and widely used clinically. INH and its metabolites, INA and SIN, are able to induce $K_{inic}$ in cells. However, INH is likely to directly induce $K_{inic}$ without the need to metabolise to INA or SIN. Many studies have shown that BNPP, a reversible carboxyesterase inhibitor, can also reduce APAP deacetylation, but the functional changes caused are unknown[47,48]. As for ABT, as a recently reported inhibitor of NAT2, although there is no direct evidence to prove the effect of BNPP on ABT, BNPP is an effective carboxyesterase inhibitor and deacetylation inhibitor. It is also very likely to affect the function of ABT, similar to APAP. Therefore, we did not do the experiment of adding APAP/ABT and BNPP together in the cells. So far, $K_{inic}$ is the only identified histone PTM with a pyridine acyl ring structure exposed to lysine. Therefore, we revealed the importance of histone $K_{inic}$ in the epigenetic regulation of chromatin structure and gene expression. Figure 6f illustrates a working model for the role of $K_{inic}$ in histone modification. In this model, INH is like to directly induces isonicotinylation of histone and non-histone proteins by generating Inic-CoA. CBP and P300 function as "writers" to add isonicotinyl group on lysines. In contrast, HDAC3 removes the isonicotinyl group as an "eraser." Once lysines on histones are isonicotinylated, the binding ability between histones and genomic DNA is weakened, thereby loosening the chromatin structure and promoting the transcription of hundreds of genes. Among them, *PIK3R1* gene expression is upregulated by $K_{inic}$, which leads to increased PIK3R1 protein levels and activates the PI3K/Akt/mTOR pathway related to HCC.

Histone acetylation with a short chain linked to lysine exhibits definite biological functions. The histone $K_{inic}$ contains a more complicated side chain linked to a lysine, and $K_{inic}$ is structurally similar to $K_{bz}$, the latter with a -CH in the benzene ring being replaced by an N atom in the pyridine ring. Despite their structural similarity, $K_{inic}$ and $K_{bz}$ as histone marks are distinct in many aspects. $K_{inic}$ and $K_{bz}$ are generated via different metabolic pathways. First, $K_{inic}$ has a pyridine acyl group, and $K_{bz}$ has an aromatic acyl group, on the lysine sidechain, and the pyridine ring and benzene ring are unique in chemical reactivities and

accessibility. For example, the hydrogenation of the pyridine ring is easier than that of the benzene ring. Second, $K_{bz}$ is stimulated by sodium benzoate, the primary component of preservatives, by increasing the concentration of benzoyl-CoA; however, $K_{inic}$ is induced by INH, an anti-tuberculosis drug, which promotes the generation of Inic-CoA. Third, lysine residues in histones modified by $K_{bz}$ primarily exist at the N-terminal tails, whereas the lysine residues in histones modified by $K_{inic}$ are widely distributed on histones from the N- to C-terminals. The discrepancy in the distribution of lysine modification sites may reflect their functional distinction in the epigenetic regulation of chromatin and gene expression. Fourth, histone $K_{inic}$ is dynamically regulated by the acetyltransferases CBP and P300 and the deacetylase HDAC3, whereas histone $K_{bz}$ is removed by deacetylase SIRT2. So far, the acyltransferase for generating $K_{bz}$ is unknown, or perhaps the acyltransferase is not required for the generation of $K_{bz}$ modification on histones. Finally, histone $K_{inic}$ is found to regulate the expression of specific genes associated with TNF, GABAergic synapse, and cAMP signalling pathways, which differ significantly from $K_{bz}$-regulated signalling pathways, such as the phospholipase D signalling pathway[8]. Hence, we anticipate that histone $K_{inic}$ may regulate more extensive biological functions than $K_{bz}$.

In this report, we determined that $K_{inic}$ exists endogenously in living cells, in both histone and non-histone proteins. Notably, we also proved the existence of Inic-CoA in cells, and it serves as a co-factor of CBP or P300 to generate $K_{inic}$ intracellularly. Exogenous INH promotes endogenous $K_{inic}$ by promoting the generation of Inic-CoA; thus, a clear metabolic map linking $K_{inic}$ with INH has been established. INH is widely used for the treatment of TB. The metabolism of INH and its role in INH-induced liver injury have been extensively studied, but the underlying mechanism remains controversial. INH itself is thought to be directly bioactivated to a reactive metabolite, which, in some patients, leads to an immune response and liver injury[49]. NAT2 is now known to be the primary enzyme that metabolises INH, and it has been shown that NAT2 deficiency increases the risk of liver injury induced by INH[50]. However, the detailed mechanisms remain unknown. In our study, when the enzymatic activity of NAT2 was inhibited, the levels of histone $K_{inic}$ increased (Fig. 2g). Intriguingly, histone $K_{inic}$ promotes the activation of the PI3K/Akt/mTOR signalling pathway by upregulating the PI3K p85 subunit. Oral feeding of INH stimulates elevated levels of isonicotinylation in the liver tissue of mice (Fig. 2f), and INH-stimulated histone $K_{inic}$ activates the PI3K/Akt/mTOR signalling pathway, regulating a variety of biological functions that contribute to the side effects of INH. Therefore, this notion is a reasonable explanation for the induction of hepatotoxicity and tumourigenicity in mice. The mechanism is also reminiscent of INH enhancing cancer risk in humans, which needs to be examined in the future for TB patients under INH treatment.

INH contains a pyridine ring structure, and $K_{inic}$ is a PTM that could bring in a pyridine structure in histone and non-histone proteins. Pyridine is widely used in the chemical and pharmaceutical industries and is also a component of some vitamins, enzymes, and various drugs, including INH. The pyridine ring-containing $K_{inic}$ histones may endow unique biological functions and play important epigenetic regulatory roles. The readers of $K_{inic}$ may be quite different from those of $K_{ac}$ and $K_{bz}$. Therefore, the biological functions regulated by $K_{inic}$ for histones and non-histone proteins should be investigated further.

$K_{inic}$ identification and characterisation represent a type of PTM on histone and non-histone proteins. However, this also raises many interesting questions. For example, where does the endogenous $K_{inic}$ come from? How is intracellular Inic-CoA synthesised? Are isonicotinylated non-histone proteins widely present in cells? Who are readers of $K_{inic}$? What can $K_{inic}$ do for

cells? Does high $K_{inic}$ cause cancer? Does $K_{inic}$ influence other PTMs on histones, for example, acetylation and methylation? These important questions warrant further investigation. In addition, the role of site-specific $K_{inic}$ on histones in the epigenetic regulation of gene transcription should be investigated thoroughly, and site-specific $K_{inic}$ antibodies need to be produced in mechanistic studies.

In summary, we demonstrated that $K_{inic}$ is a histone mark on chromatin that is metabolically regulated by INH and its metabolites. Due to the wide use of pyridine-containing chemicals in the modern world, we anticipate that histone isonicotinylation may be involved in various biological effects and diseases related to public health. This investigation may also shed light on the complicated histone code and epigenetic control of gene expression in combination with the metabolism of therapeutic drugs.

## Methods

**Chemicals and antibodies**. Unless otherwise noted, all chemical reagents were purchased from Sigma-Aldrich. Cocktail (78443) and phosphatase inhibitor cocktail (78428) were purchased from Thermo Fisher Scientific. A485 (HY-107455), SGC-CBP30 (HY-15826), Acetaminophen (APAP, HY-66005), 1-Aminobenzotriazole (ABT, HY-103389), Trichostatin A (TSA, HY-15144) and Nicotinamide (NAM, HY-B0150) were purchased from MedChemExpress. Bis-p-nitrophenyl phosphate (BNPP, N3002) was purchased from Sigma-Aldrich. D4-INH was purchased from Shanghai ZZBIO Co., Ltd., isonicotinyl-CoA was made by Shanghai Nafu Biotechnology Co., Ltd., Rabbit-pan-$K_{bz}$ (PTM-762) was purchased from PTM Biolabs: WB,1:1000. Mouse-pan-$K_{ac}$ (sc-32268) were purchased from Santa Cruz, pan-$K_{ac}$ (9441 S): IF,1:200. Rabbit-total histone H3 (9715 S), Rabbit-CBP (7389), Rabbit-P300 (86377), Rabbit-mTOR (2972 S), Rabbit-p-mTOR (Ser2448) (2971 S), Rabbit-Akt (9272 S), Rabbit-p-Akt (Ser473) (9271 T) antibodies were purchased from Cell Signaling Technology: WB,1:1000. Rabbit-anti-pan-$K_{inic}$ antibody was customer designed and prepared by PTM Biolabs Inc. (China): WB,1:1000; IF,1:200. Mouse-anti-Flag (F1804) and Rabbit-anti-HA (H3663) antibodies were purchased from Sigma-Aldrich: WB,1:2000. IF,1:400. Mouse-β-actin (66009-1-lg) and Mouse-GAPDH antibodies (60004-1-lg) were purchased from Proteintech: WB,1:3000. Mouse-anti-PIK3R1 antibody (EM1701-62) was purchased from Huabio (China): WB,1:1000. Rabbit-HDAC3 antibody (T55595) was purchased from Abmart (China): WB,1:1000. 2-step plus Poly-HRP anti-rabbit IgG detection system (PV6001) was purchased from Zhong Shan Jin Qiao, China. HiScript II Q Select RT SuperMix Kit (R223) (+gDNA wiper) and ChamQ SYBR qPCR Master Mix (Q311) were purchased from Vazyme, China.

**Dot blot analysis**. Synthetic peptides of different qualities were spotted on nitrocellulose membranes. After incubation with 5% milk for 1 h, the membranes were incubated with the first antibodies at 4 °C for 4 h, then washed three times with TBST (10 mM Tris-HCl, 150 mM NaCl, 0.1% Tween, PH = 7.5), and incubated with the secondary antibodies (anti-mouse or anti-rabbit) at room temperature for 1 h and washed three times with TBST. Then membranes were developed using enhanced chemiluminescence detection system according to protocol from the manufacturer. SageCapture Imaging Software (v2.19.12.20190311) was used for scanning.

**Cell culture**. Sf-9 cells were cultured in SIM-SF without FBS. HepG2, HeLa, HEK293T, HCT-116, HT-29, NCI-H157, NCI-H1299, MEF, HK-2, MCF-7, SUM159, SW1116, HEK293A, MCF-10A cells were cultured in DMEM with 10% FBS, 100 units/ml penicillin and 100 mg/ml streptomycin were added, and incubated at 37 °C with 5% $CO_2$.

**HPLC-MS/MS analysis and database search**. The tryptic peptides were dissolved in 0.1% formic acid (solvent A), directly loaded onto a home-made reversed-phase analytical column (15-cm length, 75 μm i.d.). The gradient was comprised of an increase from 6% to 23% solvent B (0.1% formic acid in 98% acetonitrile) over 26 min, 23% to 35% in 8 min and climbing to 80% in 3 min then holding at 80% for the last 3 min, all at a constant flow rate of 400 nL/min on an EASY-nLC 1000 UPLC system. The peptides were subjected to NSI source followed by tandem mass spectrometry (MS/MS) in Q Exactive™ Plus (Thermo) coupled online to the UPLC. The electrospray voltage applied was 2.0 kV. The m/z scan range was 350 to 1800 for full scan, and intact peptides were detected in the Orbitrap at a resolution of 70,000. Peptides were then selected for MS/MS using NCE setting as 28 and the fragments were detected in the Orbitrap at a resolution of 17,500. A data-dependent procedure that alternated between one MS scan followed by 20 MS/MS scans with 15.0 s dynamic exclusion. Automatic gain control (AGC) was set at 5E4. Fixed first mass was set as 100 m/z. The resulting MS/MS data were processed using Maxquant search engine (v.1.5.2.8). Tandem mass spectra were searched against SwissProt Human database concatenated with reverse decoy database. Trypsin/P was specified as cleavage enzyme allowing up to 4 missing cleavages. The

mass tolerance for precursor ions was set as 20 ppm in First search and 5 ppm in Main search, and the mass tolerance for fragment ions was set as 0.02 Da.

**Acid extraction of histones**. Cell pellets were lysed in hypotonic buffer (10 mM Tris–Cl pH 8.0, 1 mM KCl, 1.5 mM $MgCl_2$, and 1 mM DTT, add protease and phosphatase inhibitors before use), then resuspended in 0.4 N sulfuric acid, and the supernatants were precipitated by trichloroacetic acid. After centrifuging the histone pellets were washed with cold acetone twice and dissolved in dd$H_2O$.

**Extraction assay of total cell proteins**. Cells were collected and lysed in NP-40 buffer (50 mM Tris-HCl, 150 mM NaCl, 1% NP-40, 0.5% sodium deoxycholate, 0.1% SDS, 1 mM EDTA) supplemented with protease inhibitor cocktail and phosphatase inhibitor cocktail. A Model 680 microplate reader was used for absorbance detected in a 96 well plates for BCA protein quantification.

**Western blot analysis**. Protein extracts were separated by SDS-PAGE gel electrophoresis and transferred to PVDF membranes. After incubation with 5% milk, the membranes were incubated with the first antibodies at 4 °C overnight. The next day membranes were washed three times with TBST, then incubated with the secondary antibodies (anti-mouse or anti-rabbit) at room temperature for 1 h and washed three times with TBST. Then membranes were developed using enhanced chemiluminescence detection system according to protocol from the manufacturer. SageCapture Imaging Software (v2.19.12.20190311) was used for SDS-PAGE scanning.

**Immunofluorescent staining**. HepG2 and HeLa cells were seeded on round coverslips before experiments. The cells were washed three times with phosphate buffered saline (PBS, 137 mM NaCl, 2.7 mM KCl, 10 mM $Na_2HPO_4$, and 2 mM $KH_2PO_4$) and fixed in 4% paraformaldehyde at room temperature for 15 min. After washing with PBS three times (10 min for each time), the coverslips were incubated with 0.1% Triton X-100 at room temperature for 15 min, then blocked with 5% BSA at room temperature for 2 h, after washing with PBS three times the cells were incubated with first antibodies mixture at 4 °C overnight in wet box. The next day coverslips were washed three times with PBS, followed by incubation with secondary antibodies at room temperature for 2 h. Then, the coverslips were counterstained with 4,6-diamidino-2-phenylindole and mounted onto glass slides with nail polish. A LSM-780 confocal laser-scanning microscope (with Zeiss confocal Zen Software (v2011) using a100′/1.4 NA objective lens) was used to capture images from immuno-stained cells.

**Immunohistochemical staining**. The paraffin-embedded tissue sections were dewaxed in xylene and gradually rehydrated. Then the sections were incubated in 3% $H_2O_2$ for 30 min to quench endogenous peroxidase. To restore the antigen, slice in 10 mM sodium citrate buffer (pH = 6.0) for 20 min in 95 °C water. When incubated with pan-$K_{inic}$ antibody overnight at 4 °C, wash with PBS three times (10 min for each time), then the 2-step plus Poly-HRP anti-rabbit IgG detection system was performed. Finally, streptavidin-biotin-peroxidase method was used. All immunohistochemical staining were investigated independently by two pathologists. The assessment was classified into four grades: low reactivity marked as 1+, faint reactivity as 2+, moderate reactivity as 3+, and strong reactivity as 4+.

**Animal experiment**. Mice are on the C57BL/6 background and were maintained under a standard 12 h dark/light cycle with water and chow diet *ad libitum*. These mice (7–8 weeks old, male, $n = 10$ for each group) were oral gavage with normal saline or 50 mg/kg/day concentration of INH for ten days, and the mice were sacrificed and liver tissues were harvested for further analysis. All above studies related to animals were approved by the Peking University Health Science Center Institutional Animal Care and Use Committee.

**Patient tumour samples**. A total of ten liver cancer patients were recruited among the hospitalized patients. Patients were of both sexes at the age of 25 to 70 years. Ten paired liver cancer samples were obtained from Peking University People's Hospital from September 2018 to June 2020. All samples were obtained with informed consent at the Peking University People's Hospital. All samples were obtained with informed consent at the Peking University People's Hospital. The studies were approved by the Peking University Health Science Center Ethics Committee under IRB00001052-12088, in accordance with the declaration of Helsinki.

**In vitro isonicotinylase reaction experiment**. Transfected the HA-CBP and HA-P300 plasmids into HEK293A cells and enriched them with HA beads, then put the beads and histones extracted from HepG2 cells into HAT buffer (50 mM Tris-HCl with PH = 8.0, 50 mM NaCl, 4 mM $MgCl_2$, 0.1 mM EDTA, 1 mM DTT, 10% glycerine) at 30 °C water bath for 1 h, with the presence of 5 mM isonicotyl-CoA. Then transferred the mixture on ice for 10 min to stop the reaction. Western blot analysis was applied to detect the isonicotinylation.

**CoA extraction and HPLC-MS/MS analysis**. HepG2 cells were cultured in full media with various concentrations of INH or D4-INH treatment for 24 h, and removed the media quickly, placed the dishes on the top of dry ice. Then 2 ml 80% ethanol (v/v) was immediately added to the dishes and transferred the dishes to -80 °C freezer overnight. The next day collected the cell lysate/methanol mixture on dry ice and centrifuged at 4 °C, the supernatants were dried for HPLC-MS/MS analysis. The HPLC-MS/MS system was a 6500plus QTrap mass spectrometer coupled with ACQUITY UPLC H-Class system. An ACQUITY UPLC HSS T3 column (2.1 × 100 mm, 1.8 μm) was employed with mobile phase A: water with 5 mM ammonium acetate, and mobile phase B: methanol. Linear gradient is: 0 min, 0% B; 1.5 min, 0% B; 6 min, 95% B; 7.4 min, 95% B; 7.5 min, 0% B; and 10 min, 0% B. Flow rate was 0.3 mL/min. Column chamber and sample tray were held at 40 °C and 10 °C, respectively. Data were acquired in multiple reaction monitor (MRM) mode for Ac-CoA and INH-CoA with transitions of 873.1/428.1 in positive mode. The ion transitions were optimized using chemical standards. The nebulizer gas (Gas1), heater gas (Gas2), and curtain gas were set at 55, 55, and 30 psi, respectively. The ion spray voltage was 5500 v. The optimal probe temperature was determined to be 550 °C, and the column oven temperature was set to 40 °C. The SCIEX OS 1.6 software was applied for metabolite identification and peak integration.

**MNase sensitivity assay**. HepG2 cells were resuspended in permeabilization solution-I (150 mM sucrose, 80 mM KCl, 35 mM HEPES with pH = 7.4, 5 mM $K_2HPO_4$, 5 mM $MgCl_2$, 0.5 mM $CaCl_2$) with 0.05% Triton X-100, and incubated at room temperature for 5 min, then washed with permeabilization solution-I without Triton X-100 and resuspended in permeabilization solution-II (150 mM sucrose, 50 mM Tris-HCl with pH 7.5, 50 mM NaCl, 2 mM $CaCl_2$) containing micrococcal nuclease and incubated for 5 min at room temperature. Equal volume of 2×TNESK solution (20 mM Tris with pH 7.4, 0.2 M NaCl, 2 mM EDTA, 2% SDS, 0.2 mg/ml proteinase K) and lysis dilution buffer (150 mM NaCl, 5 mM EDTA) were added and incubated at 37 °C to stop the enzymatic reaction. Isolation of MNase digested chromatin from highly permeabilized parasite was performed using ethanol precipitation. Cold ethanol and 3 M sodium acetate (pH = 5.2) were added and incubated at -20 °C overnight. The next day after centrifuging, the pellet was washed with 80% ethanol and air dried. The chromatin was dissolved in TE buffer (pH = 8.0). Tanon GIS Software (v1.100) was used for gel scanning, and the intensity of the MNase sensitivity results was analyzed by densitometry using ImageJ software (v1.8.0).

**Real-time RT-PCR**. Total RNAs were extracted by TRIZOL reagent according to the manufacturer's protocol. Complementary DNA was prepared using HiScript II Q Select RT SuperMix Kit (+gDNA wiper) and real-time RT-PCR was performed using ChamQ SYBR qPCR Master Mix according to the manufacturer's protocol. The sequences for real-time RT-PCR primer pairs are supplied in Supplementary Table 1.

**RNA-seq analysis**. The sequencing libraries were prepared using the Next® mRNA Sample Prep Reagent Set for Illumina (NEB) as per the manufacturer's instruction. The library preparations were sequenced on an Illumina NovaSeq 6000 platform and 150 bp paired-end reads were generated. Align Reads to the Reference Genome, we took the GRch38 genome as the reference genome for this project. Sequencing reads were aligned to the reference genome sequence using Tophat2 (v2.1.1) and bowtie2 (v2.2.2) in default parameter. Genes and isoforms expression level are quantified by a software package: RSEM (RNA-Seq by Expectation Maximization v1.3.0). RSEM computes maximum likelihood abundance estimates using the Expectation-Maximization (EM) algorithm as its statistical model. EdgeR (v3.6.8) package method was used for screening differentially expressed genes. GSEA analysis of KEGG pathway database was implemented using the Diamond (v0.8.33) package.

**Statistics and reproducibility**. All data presented were repeated at least three times independently with similar results. All statistical data are presented as mean ± SEM. GraphPad Prism 8.0 (GraphPad Software Inc., La Jolla, CA) was used to plot and analyze the data, p- values were calculated by two-tailed Student's t-test and one-way ANOVA followed by Bonferrroni's multiple comparisons test. $p < 0.05$ was considered statistically significant, and marked with N.S. for no significance, * for $p < 0.5$, ** for $p < 0.01$, *** for $p < 0.001$.

**Reporting summary**. Further information on research design is available in the Nature Research Reporting Summary linked to this article.

## Data availability
The data that support this study are available from the corresponding author upon reasonable request. Mass spectrometric data generated in this study are available in the ProteomeXchange Consortium database under accession code PXD025490. RNA-seq data generated in this study have been deposited in the GEO database under accession code GSE168473. We used the GRch38 genome (accession code:GCF_000001405.39) as the reference genome for the RNA-seq analysis. Sequences for real-time RT-PCR primer pairs are supplied in Supplementary Table 1. Source data are provided with this paper.

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

## Acknowledgements

This study was supported by grants from the National Natural Science Foundation of China (grants 81730071, 81972616, 81972609, and 81773199), the Ministry of Science and Technology of China (grants 2016YFC1302103, 2015CB553906), the Beijing Natural Science Foundation (grant 7171005), Peking University (grants BMU2018JC004, BMU20120314, and PKU2021LCXQ023) to H.Z. and J.Z.

## Author contributions

Y.J. designed the research, performed experiments, analyzed the data and wrote the manuscript. Y.L. performed the animal experiments. C.L. and L.Z. performed the bioinformatic analysis. D.L. and J.Z. performed the Immunohistochemical staining. Y.W. and Z.C. analyzed the HPLC-MS/MS data. X.C. collected patient samples. H.Z. designed, supervised the research and wrote the manuscript.

## Competing interests

The authors declare no competing interests.
