## [Peer Review File · Nature Communications]

Isonicotinylation is a histone mark induced by the anti-tuberculosis first-line drug isoniazidREVIEWER COMMENTS

Reviewer #1 (Remarks to the Author):

This interesting paper report identification of lysine isonicotinylation as a new type of histone marks. The author first identified the medication by mass spectrometry, followed by various validation method. They demonstrated that this modification can be stimulated by isoniazid, an anti-tuberculosis drug. Using metabolomics, they detected sionicotinyl CoAs and further showed that this coA can be stimulated by isoniazid. They identified 26 isonicotinylation sites by mass spectrometry. Authors then carried out additional experiments to identify regulatory enzymes, demonstrated its role in epigenetics. Finally, they showed that the modification is elevated in cancer patient samples. This is the first paper that demonstrates histone acylation marks with the pyridine ring that is a component in diverse chemicals.

In general, this is an interesting, comprehensive, and novel story. The authors presented a lot of data, more than a typical paper describing a novel protein modification.

This referee most enjoy reading the paper. However, this reviewer do have some suggestions and concerns as attached below:

I would suggest to include lysine benzoylation structure in Figure 1a

The paper should be edited by a native English speakers, there are errors throughout the papers. Some examples are attached here: line 89: Enriched proteins were treated by trypsin and peptides were subjected to HPLC-MS/MS analysis; Line 208: then the histones were purified

Suggest to include the structure of INA, INH, SIN in Figure 2A

Fig 6. I would suggest to replace the structure CoA simply by CoA to make figure 6 g a little cleaner.

Fig 4e: suggest to include anti-Kac, anti_HA, and HDAC controls

I am not sure if the pan anti-Kinic works in ChIP. If it does, I would suggest to carry out ChIP-seq to check out if the histone Kinic is related with expression of genes that were described in this paper.

Reviewer #2 (Remarks to the Author):

Review: Isonicotinylation is a histone mark induced by the anti-tuberculosis first-line drug isoniazid

The authors have put together a well-designed study to investigate how the administration of isoniazid potentiates a novel post-translational modification on both histone and non-histone proteins, which could have significant biological ramifications and allows for at least the partial rationalization for isoniazid's toxicity. Therefore, the findings presented in the manuscript are significant. Below is a list of comments that the authors should address prior to resubmission.

1. Lines 84-88: The authors indicated that the study kicked off when they conducted CO-IP experiments to determine potential Kbz substrates. Did the authors consider performing a similar experiment to identify the non-histone protein targets of isonicotinylation? While the impact of the PTM on histone structure is undoubtedly important, an understanding of the subset of non-histone proteins that are modified by this PTM may give a more comprehensive understanding of the breadth of the biological significance of this modification.
2. In Fig 1a include the structure of INH for a pertinent quick reference.
3. In Fig 1b, the custom designed antibody was used to specifically detect Kinic and not Kpic or Knic. It would be interesting to provide details on the antibody and its specificity rather than simply stating that it was sourced from a vendor.
4. Lane legend for figure 2F (right) appears to be incorrect (indicates only one sample was treated with INH).
5. Figure 4B, lines 228-230: The authors state that the transfection of the acetyltransferases CBP and P300 increases isonicotinylation via immunofluorescent staining, and indeed the figure reveals the presence of these modifications. However, the figure lacks a negative control (cells transfected with empty plasmid or other appropriate control) for reference.
6. Figure 4F, lines 248-250: The authors state that cells transfected with HDAC3 have decreased isonicotinylation levels, but as with the above critique, the figure is lacking a negative control (cells transfected with empty plasmid or other appropriate control) for reference.
7. Figure 1F and 6A/B/C: For the immunohistochemistry staining images, consider adding a complementary figure that quantitatively reveals (such as bar graph) the degree of isonicotinylation for the samples.

8. In the results section on line 143, authors state that they have used a series of concentrations of INA or SIN to induce Kinic in HepG2, HEK293A and HCT-116 cells for 24 h treatment. It is evident that 10mM INH concentration was found effective and was used for further experiments. Such a high concentration of INH is never used in clinical settings and thus the results should be treated with caution. Authors should indicate and comment upon the toxicity of INH at higher concentration (reference). A High Dose of Isoniazid Disturbs Endobiotic Homeostasis in Mouse Liver Feng Li, Pengcheng Wang, Ke Liu, Mariana G. Tarrago, Jie Lu, Eduardo N. Chini, Xiaochao Ma *Drug Metab Dispos.* 2016 Nov; 44(11): 1742–1751. Published online 2016 Nov.

9. The authors claim that INH directly activates Kinic without being metabolized into INA. The direct evidence would have been addition of INH along with ABT/APAP coupled with BNPP. It would have shown that INH could not be metabolized either by NAT2 or Isoniazid hydrolase and thus the Kinic would come only and directly by INH alone. The results do not correlate with the conclusion.

10. Line -272(Results section)-Upon further analysis of Kinic the authors state that chromatin is relaxed and more accessible thus leading to upregulation of 313 genes and downregulation of 12 genes. It would have been interesting to find out if any of these genes are associated with INH treatment/metabolism. The authors should expand on this.

11. In the section 'INH-induced histone isonicotinylation activates PI3K/AKT/mTOR signaling pathway in liver cancer cells' the authors saw increased/decreased as well as no change in levels of malignant tissues as compared to normal. Thus, the conclusion that Kinic is associated with certain types of cancers and proteins needs to be reconsidered.

12. Line 129: *spodoptera frugiperda* (Genus- capital -S)

Reviewer #3 (Remarks to the Author):

In this manuscript, Jiang et al report the identification and characterization of a novel histone and non-histone modification: the lysine isonicotinylation (Kinic). This novel PTM was observed in various cell lines, mice and human tissues. They showed Kinic is induced by Isoniazid (INH), the first-line anti-tuberculosis drug, and its derivatives. Twenty-six isonicotinylation sites were identified on histones in HepG2 cells treated with 10 mM of INH for 24h by mass spectrometry. They went on to identify CBP and P300 as enzymes catalyzing histone Kinic and histone deacetylase HDAC3 as a

deisonicotinylase. They further showed that histone Kinic may relax chromatin structure and promotes gene transcription and that INH-mediated histone Kinic upregulates PIK3R1 gene expression and activates PI3K/Akt/mTOR signaling pathway in liver cancer cells, thus linking INH to the tumorigenicity in liver. Altogether, this study not only reports a new PTM with roles in transcriptional regulation but also provide a novel link to the action of first-line anti-tuberculosis drug. Overall the experiments were well designed and executed. However, to further clarify the physiological and transcriptional roles of this novel PTM, the following questions need to be addressed.

1. It is interesting that among all HDACs authors identified only HDAC3 as the deisonicotinylase. However, authors should complement their deisonicotinylase screening by loss of functional assay. In other words, if knockdown of HDAC3 but not other HDACs led to elevated histone isonicotinylation, it would be more convincing that HDAC3 is the major histone deisonicotinylase.
2. The identification of histone Kinic sites in Figure 3 was done with 10 mM of INH treated cells. What about the level and number of histone Kinic sites in the control untreated cells?
3. In Figure 5 authors analyzed the effect of INH-induced isonicotinylation on chromatin structure by MNase assay. As this assay is not very quantitative, more accurate and genomewide assay such as ATAC-seq would be a better choice to define the effect of histone Kinic on chromatin structure.
4. Authors tried to link histone isonicotinylation to transcriptional regulation. Given the availability of anti-Kinic antibody, authors should perform ChIP-seq to define Kinic genomic landscape and its relationship with gene transcription.

Reviewer #1:

We thank this reviewer for the interest of our study. This reviewer's advice and suggestions greatly improved the quality of our work. We answered all the questions literally and experimentally raised by this reviewer as following:

1. I would suggest to include lysine benzoylation structure in Figure 1a.

Response:

We thank this reviewer for the suggestion. As suggested, lysine benzoylation structure has been added in Figure 1a.

Figure 1a

a

2. The paper should be edited by a native English speaker, there are errors throughout the papers. Some examples are attached here: line 89: Enriched proteins were treated by trypsin and peptides were subjected to HPLC-MS/MS analysis; Line 208: then the histones were purified.

Response:

We thank for pointing out our errors. As suggested by this the reviewer, we have carefully edited the entire manuscript and the manuscript has been polished by a native English speaker before resubmission.

3. Suggest to include the structure of INA, INH, SIN in Figure 2A

Response:

We appreciate reviewer's suggestions to improve our manuscript. The structures of INA, INH, SIN have been added in Figure 2a.

Figure 2a

a

4. Figure 6. I would suggest to replace the structure CoA simply by CoA to make Figure 6 g a little cleaner.

Response:

Thank you again for your valuable advice, and we have replaced the structure of Inic-CoA in the Figure 6f (we changed Figure 6g to Figure 6f) to make it clear and accurate.

Figure 6f

f

5. Figure 4e: suggest to include anti-K_{ac}, anti-HA, and HDAC controls.

Response:

We are so grateful for this referee's critical comments which are very helpful for us.

As suggested, we have added the anti-K_{ac}, anti-HA, anti-HDAC also anti-β-actin

Western blot results in Figure 4f (we have changed Figure 4e to Figure 4f).

Figure 4f

6. I am not sure if the pan anti-K_{inic} works in ChIP. If it does, I would suggest to carry out ChIP-seq to check out if the histone K_{inic} is related with expression of genes that were described in this paper.

Response:

Thank you for this very important suggestion. As you mentioned, we initially expected to be able to conduct a dual analysis of both the CHIP-seq and RNA-seq. However, the pan anti-K_{inic} antibody did not work so well in the CHIP-seq analysis, and it is a pity that we failed to perform CHIP-seq to find the direct genes that are regulated by isonicotinylation. In the near future we will continue to pursuit the CHIP-seq analysis by improving the quality pan anti-K_{inic} antibody. The negative sequencing results of CHIP-seq using the present pan anti-K_{inic} antibody are shown as below:

The left picture shows the DNA distribution of untreated HepG2 cells after pan- K_{inic} antibody enrichment and fragmentation, and the right picture shows the DNA distribution of INH-treated HepG2 cells after pan- K_{inic} antibody enrichment and fragmentation. As shown above, the two samples did not have well enriched DNA.

Reviewer #2:

We greatly appreciated this reviewer for the comments and suggestions that greatly improved the quality of our manuscript and enlarged our understanding in this area. According to the advice and ideas from this reviewer all the suggested experiments have been thoroughly completed. We have listed them and answered the questions as following:

1. Lines 84-88: The authors indicated that the study kicked off when they conducted CO-IP experiments to determine potential K_{bz} substrates. Did the authors consider performing a similar experiment to identify the non-histone protein targets of isonicotinylation? While the impact of the PTM on histone structure is undoubtedly important, an understanding of the subset of non-histone proteins that are modified by this PTM may give a more comprehensive understanding of the breadth of the biological significance of this modification.

Response:

We thank this reviewer for the very constructive suggestion. As you suggested, we supplied in the Supplementary Figure 3, in which we also identified that many non-histone proteins undergo isonicotinylation after INH-treatment by western blot assay. The PTMs of non-histone proteins have many functions and mechanisms. We will continue to investigate the biological role of non-histone protein isonicotinylation in the future investigation.

2. In Figure 1a include the structure of INH for a pertinent quick reference.

Response:

We appreciate this reviewer's suggestion. The structures of INA, INH, SIN have been added in Figure 2a.

Figure 2a

3. In Figure 1b, the custom designed antibody was used to specifically detect K_{inic} and not K_{pic} or K_{nic} . It would be interesting to provide details on the antibody and its specificity rather than simply stating that it was sourced from a vendor.

Response:

Thanks to this reviewer for the very suggestion. We showed in the Figure 1b that the pan- K_{inic} antibody prepared has good specificity.

Figure 1b

The process of preparing this pan- K_{inic} antibody is as follows:

1. First, we designed and synthesized two antigenic peptides for isonicotinylation according to the modified amino acid sequence for animal immunization, purification and detection. Then four SPF experimental-grade New Zealand white rabbits were immunized several times, and the four rabbits were named as R9, R10, R11, R12, but the R12 rabbit died after the third immunization. In a 96-well ELISA plate coated with antigens: modified peptides or non-modified control peptides, add antiserum to incubate at different dilutions (for example, 1:54K, 1:162K). Afterwards, an enzyme-

labeled secondary antibody and a chemiluminescent substrate are applied, and the combination of the polypeptide and the antiserum is detected by a chemical development assay. According to empirical values, when the absorbance at 450nm (OD450) wavelength is approximately equal to 1, the corresponding dilution ratio of the antiserum is the titer of the antiserum. The data whose OD450 value is approximately equal to 1 is marked with gray shading. In general, the antiserum of the three rabbits all showed well titer, and the signal for identifying isonicotinylated peptides was about 10 times stronger than other modified peptides (nicotinylation, picolinylated, benzoylation).

R9								
	Pan nicotinyllysine Peptide 1	Pan nicotinyllysine Peptide 2	Pan picolinyllysine Peptide 3	Pan picolinyllysine Peptide 4	Pan isonicotinyllysine Peptide 5	Pan isonicotinyllysine Peptide 6	Pan benzoyllysine Peptide 7	Control Peptide 8
1:18K	0.269	0.953	0.288	1.307	2.088	2.285	0.634	0.043
1:54K	0.131	0.441	0.139	0.578	1.42	1.67	0.306	0.058
1:162K	0.069	0.182	0.091	0.226	0.657	0.804	0.165	0.051
1:486K	0.06	0.095	0.058	0.107	0.283	0.321	0.116	0.054
1:1458K	0.053	0.06	0.05	0.063	0.147	0.136	0.103	0.049
R10								
	Pan nicotinyllysine Peptide 1	Pan nicotinyllysine Peptide 2	Pan picolinyllysine Peptide 3	Pan picolinyllysine Peptide 4	Pan isonicotinyllysine Peptide 5	Pan isonicotinyllysine Peptide 6	Pan benzoyllysine Peptide 7	Control Peptide 8
1:18K	1.399	2.548	1.324	2.421	2.672	2.533	0.265	0.192
1:54K	0.65	1.952	0.698	1.56	2.454	2.584	0.273	0.195
1:162K	0.257	1.04	0.39	0.739	2.108	1.824	0.265	0.221
1:486K	0.313	0.823	0.305	0.458	1.116	0.922	0.254	0.172
1:1458K	0.13	0.442	0.246	0.299	0.574	0.437	0.224	0.05
R11								
	Pan nicotinyllysine Peptide 1	Pan nicotinyllysine Peptide 2	Pan picolinyllysine Peptide 3	Pan picolinyllysine Peptide 4	Pan isonicotinyllysine Peptide 5	Pan isonicotinyllysine Peptide 6	Pan benzoyllysine Peptide 7	Control Peptide 8
1:18K	0.24	0.556	0.197	0.647	0.673	1.42	0.14	0.061
1:54K	0.107	0.23	0.103	0.263	0.28	0.625	0.13	0.061
1:162K	0.064	0.103	0.065	0.121	0.157	0.259	0.135	0.113
1:486K	0.057	0.071	0.054	0.071	0.102	0.105	0.115	0.055
1:1458K	0.051	0.055	0.052	0.055	0.089	0.067	0.128	0.053

2. Take sufficient serum of rabbit R9, R10, R11 for multi-step affinity purification.

The purified antibodies are tested for quality using ELISA, Dot Blot and WB experiments.

(1) In a 96-well microtiter plate coated with antigen-modified peptides or other modified peptides with similar structures, add antibodies to incubate in different dilution ratios (for example, 1:54K, 1:162K). Afterwards, an enzyme-labeled secondary antibody and a chemiluminescent substrate are applied, and the binding of the polypeptide and the antibody is detected by a chemical development assay. R9, R10, and R11 correspond to the purified antibody identification of rabbit serum as Ab9-1, Ab9-2, Ab10-1, Ab10-2, Ab11. The ELISA results showed:

The five antibodies of Ab9-1, Ab9-2, Ab10-1, Ab10-2, and Ab11 recognized the titers of isonicotinylated peptides greater than 1:54K. Ab9-1, Ab9-2, Ab10-1, Ab10-2 recognize isonicotinylated polypeptide signals 10 times stronger than the other modified polypeptides (nicotinylation, picolinylation, benzoylation). Ab11 antibody recognizes picolinylated peptides.

Antibody purified from R9 (Ab9-1)							
	Pan nicotinyllysine Peptide 1	Pan nicotinyllysine Peptide 2	Pan picolinyllysine Peptide 3	Pan picolinyllysine Peptide 4	Pan isonicotinyllysine Peptide 5	Pan isonicotinyllysine Peptide 6	Pan benzoyllysine Peptide 7
1:18K	0.171	0.733	0.223	0.497	1.68	1.842	0.795
1:54K	0.117	0.421	0.133	0.252	1.599	1.677	0.394
1:162K	0.1	0.203	0.088	0.119	1.177	1.411	0.173
1:486K	0.083	0.126	0.079	0.092	0.687	0.883	0.11
1:1458K	0.072	0.087	0.076	0.072	0.308	0.426	0.084

Antibody purified from R9 (Ab9-2)							
	Pan nicotinyllysine Peptide 1	Pan nicotinyllysine Peptide 2	Pan picolinyllysine Peptide 3	Pan picolinyllysine Peptide 4	Pan isonicotinyllysine Peptide 5	Pan isonicotinyllysine Peptide 6	Pan benzoyllysine Peptide 7
1:18K	0.197	0.849	0.261	0.973	1.652	1.645	0.95
1:54K	0.133	0.478	0.153	0.503	1.475	1.588	0.545
1:162K	0.086	0.223	0.096	0.231	1.061	1.268	0.251
1:486K	0.084	0.12	0.083	0.129	0.61	0.861	0.135
1:1458K	0.079	0.089	0.076	0.089	0.282	0.381	0.09

Antibody purified from R10 (Ab10-1)							
	Pan nicotinyllysine Peptide 1	Pan nicotinyllysine Peptide 2	Pan picolinyllysine Peptide 3	Pan picolinyllysine Peptide 4	Pan isonicotinyllysine Peptide 5	Pan isonicotinyllysine Peptide 6	Pan benzoyllysine Peptide 7
1:18K	0.26	1.13	0.388	0.929	1.796	1.789	0.159
1:54K	0.143	0.855	0.328	0.464	1.789	1.73	0.104
1:162K	0.098	0.616	0.18	0.206	1.485	1.498	0.078
1:486K	0.076	0.362	0.136	0.121	1.118	1.046	0.075
1:1458K	0.076	0.191	0.067	0.077	0.634	0.533	0.07

Antibody purified from R10 (Ab10-2)							
	Pan nicotinyllysine Peptide 1	Pan nicotinyllysine Peptide 2	Pan picolinyllysine Peptide 3	Pan picolinyllysine Peptide 4	Pan isonicotinyllysine Peptide 5	Pan isonicotinyllysine Peptide 6	Pan benzoyllysine Peptide 7
1:18K	0.248	1.105	0.06	0.321	1.809	1.842	0.063
1:54K	0.16	0.94	0.073	0.165	1.656	1.653	0.075
1:162K	0.106	0.661	0.065	0.098	1.429	1.416	0.063
1:486K	0.082	0.355	0.076	0.079	0.985	0.9	0.067
1:1458K	0.076	0.189	0.071	0.075	0.544	0.411	0.066

Antibody purified from R11 (Ab11)							
	Pan nicotinyllysine Peptide 1	Pan nicotinyllysine Peptide 2	Pan picolinyllysine Peptide 3	Pan picolinyllysine Peptide 4	Pan isonicotinyllysine Peptide 5	Pan isonicotinyllysine Peptide 6	Pan benzoyllysine Peptide 7
1:18K	0.181	0.64	0.129	1.544	1.422	1.796	0.428
1:54K	0.101	0.324	0.096	1.15	1.042	1.597	0.219
1:162K	0.075	0.152	0.077	0.625	0.543	1.139	0.111
1:486K	0.073	0.111	0.077	0.259	0.255	0.612	0.087
1:1458K	0.071	0.077	0.07	0.148	0.127	0.272	0.073

(2) Dot blot experiments for antibodies validation: Different doses (for example, 4 ng, 20 ng) of antigen-modified peptides and other structurally similar modified peptides are immobilized on NC membranes, antibodies are applied for incubation, and enzyme-labeled secondary antibodies and chemiluminescent substrates are then applied to detect the binding of the polypeptides and the antibodies. The results

showed that the Dot blot detection limit of Ab9-1, Ab10-1, Ab10-2 reached 4 ng, and the Dot blot detection limit of Ab9-2 reaches 20 ng, these four antibodies basically do not recognize other modified peptides. The detection limit of Ab11 antibody Dot blot reached 100 ng, which did not meet the requirements.

(3) Western blot experiments for antibodies validation: The total protein of Neuro-2a (N2a) from mouse neuroma blasts was extracted for WB detection. The results showed that Ab9-1, Ab9-2, Ab10-2, Ab11 four antibodies can detect more bands.

Based on the above experimental results, we chose Ab9-1, Ab9-2, Ab10-2 as the antibody for subsequent experiments.

4. Lane legend for Figure 2F (right) appears to be incorrect (indicates only one sample was treated with INH).

Response:

Thank you for pointing out our error. We have corrected this error in the revised manuscript (we changed Figure 2f to Figure 2g).

Figure 2g

g

5. Figure 4B, lines 228-230: The authors state that the transfection of the acetyltransferases CBP and P300 increases isonicotinylation via immunofluorescent staining, and indeed the Figure reveals the presence of these modifications. However, the Figure lacks a negative control (cells transfected with empty plasmid or other appropriate control) for reference.

Response:

We thank this reviewer for this very important advice. According to the advice, in

Figure 2b and Supplementary Figure 4a, we have added samples of HepG2 or HeLa cells transfected with empty plasmid 3×HA, and levels of K_{ac} and K_{inic} were observed. The immunofluorescence results showed that after transfecting the empty plasmid 3×HA, the K_{ac} and K_{inic} levels were not be affected.

Figure 2b

Supplementary Figure 4a

6. Figure 4F, lines 248-250: The authors state that cells transfected with HDAC3 have decreased isonicotylation levels, but as with the above critique, the Figure is lacking a negative control (cells transfected with empty plasmid or other appropriate control) for reference.

Response:

We thank this reviewer for this very important advice. According to the advice, in Figure 4e (we changed Figure 4f to Figure 4e) and Supplementary Figure 4d, we added samples of HepG2 cells transfected with empty plasmid 3×Flag, and levels of

K_{ac} and K_{inic} were observed. The immunofluorescence results showed that after transfecting the empty plasmid 3×HA, the K_{ac} and K_{inic} levels were not affected.

Figure 4e

Supplementary Figure 4d

7. Figure 1F and 6A/B/C: For the immunohistochemistry staining images, consider adding a complementary Figure that quantitatively reveals (such as bar graph)

the degree of isonicotinylation for the samples.

Response:

We thank this reviewer for this very good idea. In Figure 1f, we performed immunohistochemical experiments on the normal thyroid, esophageal epithelium, fundus glands, colon and lung tissues, and the assessment was classified into four grades: low reactivity marked as 1+, faint reactivity as 2+, moderate reactivity as 3+, and strong reactivity as 4+, data were presented as mean \pm s.d. .We found that in the normal thyroid, colon and lung tissues, the K_{inc} levels are lower, however, in the normal esophageal epithelium and fundic gland tissues, the K_{inc} levels are higher. In addition, we noticed an interesting phenomenon, in the normal esophageal epithelium tissues, the isonicotinylation modification of the well-differentiated outer layer is higher (Figure 1f).

Figure 1f

Further, in Figure 6a & b and Supplementary Figure 5a, we applied immunohistochemical staining to determine the levels of isonicotinylation in a variety of tumors and normal tissues. Results showed that the level of total K_{inic} in gastric carcinoma tissues was decreased than that of in normal tissues; and the levels of total K_{inic} in lung adenocarcinoma and thyroid carcinoma tissues were increased than that of in normal tissues. However, the levels of total K_{inic} in renal carcinoma, rectal carcinoma, colonic adenocarcinoma, mammary adenocarcinoma, pancreatic carcinoma, esophageal carcinoma and hepatocellular carcinoma tissues showed no significant difference from that of in normal tissues. Data were presented as mean \pm s.d.. For a single comparison of 2 groups, Student's t test was performed. Values with $*P < 0.05$, $**P < 0.01$ are considered significant.

Figure 6a&b

Supplementary Figure 5a

a

Renal carcinoma

Normal adjacent tumor

Tumor

Rectal carcinoma

Normal adjacent tumor

Tumor

Colonic adenocarcinoma

Normal adjacent tumor

Tumor

Mammary adenocarcinoma

Normal adjacent tumor

Tumor

Pancreatic carcinoma

Normal adjacent tumor

Tumor

Esophageal carcinoma

Normal adjacent tumor

Tumor

Hepatocellular carcinoma

Normal adjacent tumor

Tumor

8. In the results section on line 143, authors state that they have used a series of concentrations of INA or SIN to induce K_{inic} in HepG2, HEK293A and HCT-116 cells for 24 h treatment. It is evident that 10mM INH concentration was found effective and was used for further experiments. Such a high concentration of INH is never used in clinical settings and thus the results should be treated with caution. Authors should indicate and comment upon the toxicity of INH at higher concentration (reference). A High Dose of Isoniazid Disturbs Endobiotic Homeostasis in Mouse Liver. Feng Li, Pengcheng Wang, Ke Liu, Mariana G. Tarrago, Jie Lu, Eduardo N. Chini, Xiaochao Ma. *Drug Metab Dispos.* 2016 Nov; 44(11): 1742–1751. Published online 2016 Nov.

Response:

We thank this reviewer for this very important comment and suggestion. The concentration of isoniazid for the treatment of cell lines is 10 mM, and our reasons are as follows:

1. Due to previous reports on the PTMs of histones, the concentration of drug treatment for cells is at the range of 5-25 mM^[1, 2, 3], we therefore chose 10 mM as the treatment concentration.
2. The intragastric concentration of isoniazid in mice is 50 mg/kg/day, since the regular dose of INH is 5 mg/kg/day in human for TB (tuberculosis) therapy and 50 mg/kg/day INH in mouse mimics the pharmacological dose in human^[4]. It turns out that it can also induce the K_{inic} of histones and non-histone proteins, which proves the low concentrations of isoniazid drug treatment can also induce isonicotinylation in animals, instead of only high concentrations (Please refer to Figure 2f).
3. Because the titer of our pan- K_{inic} antibody is not very high, a higher concentration of isoniazid is required to induce obvious K_{inic} in the histones.
4. Through the experimental results observed after treatment with 10 mM isoniazid, the function and mechanism of isoniazid-induced K_{inic} modification can be observed very intuitively and strongly.

And thank you for your recommendation and we quoted the article you suggested.

- [1] Huang H, Zhang D, Wang Y, Perez-Neut M, Han Z, Zheng YG, Hao Q and Zhao Y. Lysine benzoylation is a histone mark regulated by SIRT2. *Nature Communications*. 2018,9(1).
- [2] Zhang D, Tang Z, Huang H, Zhou G, Cui C, Weng Y, Liu W, Kim S, Lee S, Perez-Neut M, Ding J, Cxyz D, Hu R, Ye Z, He M, Zheng YG, Shuman HA, Dai L, Ren B, Roeder RG, Becker L and Zhao Y. Metabolic regulation of gene expression by histone lactylation. *Nature*. 2019 Oct,574(7779):575-580.
- [3] Sabari BR, Tang Z, Huang H, Yong-Gonzalez V, Molina H, Kong HE, Dai L, Shimada M, Cross JR, Zhao Y, Roeder RG and Allis CD. Intracellular Crotonyl-CoA Stimulates Transcription through p300-Catalyzed Histone Crotonylation. *Mol Cell*. 2018 Feb 1,69(3):533.
- [4] Li F, Wang P, Liu K, Tarrago MG, Lu J, Chini EN and Ma X. A High Dose of Isoniazid Disturbs Endobiotic Homeostasis in Mouse Liver. *Drug metabolism and disposition: the biological fate of chemicals*. 2016 Nov,44(11):1742-1751.

9. The authors claim that INH directly activates K_{inic} without being metabolized into INA. The direct evidence would have been addition of INH along with ABT/APAP coupled with BNPP. It would have shown that INH could not be metabolized either by NAT2 or Isoniazid hydrolase and thus the K_{inic} would come only and directly by INH alone. The results do not correlate with the conclusion.

Response:

We thank the reviewer for pointing out this issue. This issue, as you mentioned, when we initially designed the experiment, we also thought that INH should be added together with APAP/ABT and BNPP. But it is worth noting that there are several articles have proved that BNPP, a reversible carboxyesterase inhibitor, also reduce APAP deacetylation^[5, 6], and the caused functional changes is unknown. As for ABT, as a recently reported inhibitor of NAT2, although there is no direct evidence to prove the effect of BNPP on it, BNPP as an effective carboxyesterase inhibitor and deacetylation inhibitor, it is also very likely to affect the function of ABT like APAP. Therefore, we did not do the experiment of adding APAP/ABT and BNPP together in the cells.

And we thought that when ABT/APAP/BNPP work alone, the conversion of INH to INA is inhibited, but we observed that the level of histone K_{inic} modification did not decrease, but increased slightly. This is actually consistent with the phenomenon

observed in our previous experiments (the isoniazid-treated cells have more obvious changes in isonicotinylation level compared to isonicotinic acid).

Thus, we toned down the conclusion to “INH is likely to induce histone K_{inic} directly, without requirement of INH turnover into INA” instead of “INH induces histone K_{inic} directly, without requirement of INH turnover into INA”.

- [5] Newton JF, Kuo CH, DeShone GM, Hoefle D, Bernstein J and Hook JB. The role of p-aminophenol in acetaminophen-induced nephrotoxicity: effect of bis(p-nitrophenyl) phosphate on acetaminophen and p-aminophenol nephrotoxicity and metabolism in Fischer 344 rats. *Toxicol Appl Pharmacol.* 1985 Dec,81(3 Pt 1):416-430.
- [6] Mugford CA and Tarloff JB. Contribution of oxidation and deacetylation to the bioactivation of acetaminophen in vitro in liver and kidney from male and female Sprague-Dawley rats. *Drug metabolism and disposition: the biological fate of chemicals.* 1995 Feb,23(2):290-294.

10. Line -272(Results section)-Upon further analysis of K_{inic} the authors state that chromatin is relaxed and more accessible thus leading to upregulation of 313 genes and downregulation of 121 genes. It would have been interesting to find out if any of these genes are associated with INH treatment/metabolism. The authors should expand on this.

Response:

We greatly appreciated the idea from this reviewer. At present, there are very few reports on gene changes about INH treatment/metabolism. We only get two genes (NAT2 and CYP2E1) from the website <https://www.genome.jp/pathway/hsa00983> and related publications^[7, 8], we performed an enrichment analysis between them and our RNA-seq results, and found that the seq results are not significantly correlated with these two genes. However, we will continue to pursuit to clarify the genes that

are associated with INH treatment/ metabolism in the future investigations.

- [7] Hassan HM, Guo HL, Yousef BA, Luyong Z and Zhenzhou J. Hepatotoxicity mechanisms of isoniazid: A mini-review. Journal of applied toxicology : JAT. 2015 Dec,35(12):1427-1432.
- [8] Liu K, Li F, Lu J, Gao Z, Klaassen CD and Ma X. Role of CYP3A in isoniazid metabolism in vivo. Drug metabolism and pharmacokinetics. 2014,29(2):219-222.

11. In the section 'INH-induced histone isonicotinylation activates PI3K/AKT/mTOR signaling pathway in liver cancer cells the authors saw increased/decreased as well as no change in levels of malignant tissues as compared to normal. Thus, the conclusion that K_{inic} is associated with certain types of cancers and proteins needs to be reconsidered.

Response:

We thank this reviewer for the good suggestion. Yes, INH-induced activation of PI3K/AKT/mTOR signaling pathway may be caused by histone isonicotinylation or non-histone isonicotinylation, which are correlated with tumor development. We,

therefore, tone down the section title to “INH-induced histone isonicotinylation varies in tumors and upregulated cancer-related signalling pathway”.

12. Line 129: *spodoptera frugiperda* (Genus- capital -S)

Response:

Thank you very much for advice. We have changed this improper expression to “insect cells Sf-9 (*Spodoptera frugiperda* 9)” in the manuscript.

Reviewer #3:

We greatly appreciated your important comments and suggestions for our manuscript. We have improved our manuscript substantially under your advice. The suggested experiments have been thoroughly followed and completed. We listed all of them one by one as following:

1. It is interesting that among all HDACs authors identified only HDAC3 as the deisonicotinylase. However, authors should complement their deisonicotinylase screening by loss of functional assay. In other words, if knockdown of HDAC3 but not other HDACs led to elevated histone isonicotinylation, it would be more convincing that HDAC3 is the major histone deisonicotinylase.

Response:

We are so grateful for this reviewer’s critical comments which are very helpful for us. As suggested, we did the experiment and have supplemented the result of this experiment in Supplementary Figure 4f. For detail, HepG2 cells were transfected with two small interfering RNAs (siRNAs) against HDAC3, then core histones were acid-extracted and tested using pan- K_{inic} , pan- K_{ac} and H3 antibodies. Total histones were visualized with Coomassie blue staining. Total proteins were extracted and tested using HDAC3 and β -actin antibodies by Western blot analysis. Indeed, both K_{inic} and

K_{ac} were increased upon depletion of HDAC3, suggesting that HDAC3 is the deisonicotinylase. However, we still remain open for other HDAC may also function as deisonicotinylases.

Supplementary Figure 4f

2. The identification of histone Kinic sites in Figure 3 was done with 10 mM of INH treated cells. What about the level and number of histone Kinic sites in the control untreated cells?

Response:

We thank this reviewer for this very important comment. After treating cells with 10 mM isoniazid, we have detected 26 histone lysine isonicotinylation sites by mass spectrometry. As for the isonicotinylation sites in the control untreated cells, we did not obtain the sites information. The reasons we thought is as following:

In our Figure 1, we have proved that the isonicotinylation is endogenously existed in cells without isoniazid treatment by using mass spectrometry. However, the endogenous isonicotinylation is very weak without stimulation by isoniazid and could not be adequately enriched by the pan-K_{inic} antibody, and followed by detection using

mass spectrometry or Western blot analysis. Taken together, the endogenous histone K_{inic} sites were undetectable without isoniazid treatment.

3. In Figure 5 authors analyzed the effect of INH-induced isonicotinylation on chromatin structure by MNase assay. As this assay is not very quantitative, more accurate and genomewide assay such as ATAC-seq would be a better choice to define the effect of histone K_{inic} on chromatin structure.

Response:

We greatly appreciated for the comment from this reviewer. As you mentioned, at the beginning we looked forward to performing ATAC-seq sequencing analysis, RNA-seq analysis and chip-seq analysis for a joint analysis to support our statement. We first performed the RNA-seq sequencing experiment, but unfortunately due to the impact of COVID-19 at that time, our experiments were affected, it turned out to be that the ATAC-seq sequencing sample and the RNA-seq sequencing sample were not being the same batch of HepG2 cells. Perhaps due to this reason, the overlap between our ATAC-seq and RNA-seq is not good, only one gene overlapped. Therefore, we took the classic MNase sensitivity experiment and the detection of the heterochromatin marker gene *Sat-2* to examine the histone K_{inic} -promoted chromatin accessibility. In the Figure below, the right filled circle represents the elevated genes enriched in the RNA-seq result, and the left filled circle represents the open genes enriched in the ATAC-seq result, showing that only one gene is overlapped. Apparently, this result is poor and we will continue to work on the ATAC-seq to define the effect of histone K_{inic} on chromatin accessibility.

4. Authors tried to link histone isonicotinylation to transcriptional regulation. Given the availability of anti-Kinic antibody, authors should perform CHIP-seq to define Kinic genomic landscape and its relationship with gene transcription.

Response:

Indeed, this is a very good suggestion. As you mentioned, we initially expected to be able to conduct a dual analysis of both the CHIP-seq and RNA-seq. However, the pan anti-Kinic antibody did not work so well in the CHIP-seq analysis, and it is a pity that we failed to perform CHIP-seq to find the direct genes that are regulated by isonicotinylation. In the near future we will continue to pursue the CHIP-seq analysis by improving the quality pan anti-K_{inic} antibody. The negative sequencing results of CHIP-seq using the present pan anti-K_{inic} antibody are shown as below:

The left picture shows the DNA distribution of untreated HepG2 cells after pan- K_{inic} antibody enrichment and fragmentation, and the right picture shows the DNA distribution of INH-treated HepG2 cells after pan- K_{inic} antibody enrichment and fragmentation. As shown above, the two samples did not have well enriched DNA.

REVIEWERS' COMMENTS

Reviewer #2 (Remarks to the Author):

The authors have addressed all of our suggestions and the paper should be accepted.

Reviewer #3 (Remarks to the Author):

They have to large extent explain or address my questions. I have no further question.